# Structural and functional analysis of the GABARAP interaction motif (GIM)

Vladimir V Rogov[1,†], Alexandra Stolz[2,†], Arvind C Ravichandran[3,†], Diana O Rios-Szwed[4], Hironori Suzuki[3,5], Andreas Kniss[1], Frank Löhr[1], Soichi Wakatsuki[5,6,7], Volker Dötsch[1], Ivan Dikic[2,8,*] [ID], Renwick CJ Dobson[3,9,**] [ID] & David G McEwan[2,4,***] [ID]

## Abstract

Through the canonical LC3 interaction motif (LIR), [W/F/Y]-$X_1$-$X_2$-[I/L/V], protein complexes are recruited to autophagosomes to perform their functions as either autophagy adaptors or receptors. How these adaptors/receptors selectively interact with either LC3 or GABARAP families remains unclear. Herein, we determine the range of selectivity of 30 known core LIR motifs towards individual LC3s and GABARAPs. From these, we define a GABARAP Interaction Motif (GIM) sequence ([W/F]-[V/I]-$X_2$-V) that the adaptor protein PLEKHM1 tightly conforms to. Using biophysical and structural approaches, we show that the PLEKHM1-LIR is indeed 11-fold more specific for GABARAP than LC3B. Selective mutation of the $X_1$ and $X_2$ positions either completely abolished the interaction with all LC3 and GABARAPs or increased PLEKHM1-GIM selectivity 20-fold towards LC3B. Finally, we show that conversion of p62/SQSTM1, FUNDC1 and FIP200 LIRs into our newly defined GIM, by introducing two valine residues, enhances their interaction with endogenous GABARAP over LC3B. The identification of a GABARAP-specific interaction motif will aid the identification and characterization of the expanding array of autophagy receptor and adaptor proteins and their *in vivo* functions.

**Keywords** autophagy; Atg8; GABARAP; LC3; PLEKHM1
**Subject Categories** Autophagy & Cell Death; Structural Biology

## Introduction

Autophagy is an alternative catabolic process that works alongside the proteasome for the degradation of cellular material. Such cargo can include protein aggregates, damaged organelles, intracellular pathogens, metabolic substrates and ferritin aggregates [1–4]. At the heart of the autophagy pathway are ubiquitin-like proteins that, despite sharing little primary sequence with ubiquitin, contain an ubiquitin-like fold [5]. Best characterized upon these ubiquitin-like modifiers is the *Saccharomyces cerevisiae* Atg8 protein. Unlike *Saccharomyces cerevisiae*, however, there are six Atg8 homologues in mammals (mammalian Atg8s; mATG8s) that, presumably, have distinct or overlapping functions: MAP1LC3A (microtubule-associated protein light chain 3 alpha; LC3A), LC3B, LC3C, GABARAP (γ-aminobutyric acid receptor-associated protein), GABARAP-L1 and GABARAP-L2/GATE-16 [6].

All six mATG8s are essential for autophagy, are conjugated to autophagosomes and serve to recruit two broad classes of molecules: autophagy receptors and autophagy adaptors. Autophagy receptors interact directly with mATG8s on the inner autophagosomal membrane and provide a vital link between the autophagosomal isolation membrane and cargo to be sequestered and delivered to the lysosome for degradation, for example, protein aggregates (p62 [7]; NBR1 [8]; Cue5 [9]) or intracellular pathogens (OPTN [10]; NDP52 [11]; TAX1BP1 [12]). Additionally, organelles, such as ER (FAM134B [4]), mitochondria (Nix/BNIP3L [13]; FUNDC1[14]) as well as ferritin (NCOA4 [15]), can be specifically targeted by autophagy receptors. On the other hand, autophagy adaptor proteins interact with mATG8 proteins on the convex autophagosomal membrane surface and can regulate autophagosome formation (ULK1/2 [16]), autophagosome transport (FYCO1

1  Institute of Biophysical Chemistry and Center for Biomolecular Magnetic Resonance, Goethe University, Frankfurt am Main, Germany
2  Institute of Biochemistry II, Goethe University School of Medicine, Frankfurt (Main), Germany
3  Biomolecular Interaction Centre, School of Biological Sciences, University of Canterbury, Christchurch, New Zealand
4  Division of Cell Signalling & Immunology, School of Life Sciences, University of Dundee, Dundee, UK
5  Structural Biology Research Centre, Photon Factory, Institute of Materials Structure Science, High Energy Accelerator Research Organization (KEK), Tsukuba, Ibaraki, Japan
6  Photon Science, SLAC National Accelerator Laboratory, Menlo Park, CA, USA
7  Structural Biology (School of Medicine), Beckman Center B105, Stanford, CA, USA
8  Buchmann Institute for Molecular Life Sciences, Goethe University, Frankfurt am Main, Germany
9  Department of Biochemistry and Molecular Biology, Bio21 Institute, University of Melbourne, Parkville, Vic., Australia
   *Corresponding author. Tel: +49 69 6301 4546; E-mail: ivan.dikic@biochem2.de
   **Corresponding author. Tel: +64 3 369 5145 ext 95145; E-mail: renwick.dobson@canterbury.ac.nz
   ***Corresponding author. Tel: +44 1382 386337; E-mail: d.g.mcewan@dundee.ac.uk
   †These authors contributed equally to this work

   

[17]), crosstalk with the endocytic network (TBC1D5 [18]) and autophagosome fusion with the lysosome (PLEKHM1 [19]), but are themselves not degraded by autophagy. Autophagy ubiquitin-like modifiers can also act as signalling scaffolds to attract diverse complexes, such as GABARAP-mediated recruitment of CUL3-KBTBD6/KBTBD7 ubiquitin ligase complex to a membrane-localized substrate, TIAM1 [20]. One essential common feature of all adaptors and receptors is the presence of a LC3 interaction region [LIR; also known as LC3 interaction motif (LIM) or Atg8 interaction motif (AIM)].

With some known exceptions ("atypical LIRs/LIMs"), such as NDP52 [11], TAX1BP1 [21] and the dual LIR/UFIM (UFM1-Interaction Motif) in UBA5 [22], the majority of LIRs contain a core $\Theta$-$X_1$-$X_2$-$\Gamma$ motif, where $\Theta$ is an aromatic residue (W/F/Y) and $\Gamma$ is a large hydrophobic residue (L/V/I). Structural studies have shown that the side chains of the aromatic residue ($\Theta$) within the core LIR motif are placed deep inside of a hydrophobic pocket (HP1) on the Atg8/LC3/GABARAP surface, formed between $\alpha$-helix 2 and $\beta$-strand 2, while side chains of the hydrophobic LIR residues ($\Gamma$) occupies a second hydrophobic pocket (HP2) between $\beta$-strand 2 and $\alpha$-helix 3 (reviewed in [3,23,24]). Acidic and phosphorylatable serine/threonine residues N-terminal, and occasionally C-terminal, to the core LIR/AIM can contribute to the stabilization of LIR–mATG8 interactions [25–27].

There is growing evidence that the function of the autophagy adaptors and receptors are closely linked to their interaction with specific LC3/GABARAP family members and their distinct role in the pathway [19,28,29]. The presence of six similar LC3/GABARAP proteins also points towards their specific functions within the pathway; for example, at the formation and closure of the nascent phagophore during autophagosome formation [29]. Therefore, despite having similar sequences, there is a clear selectivity and divergence of function between the six mATG8s. However, as yet, there has been no identification of an LC3 or GABARAP subfamily-selective LIR motif.

In order to address the issue of selectivity, we implemented a peptide-based assay to screen 30 validated LIR sequences against all LC3 and GABARAP proteins, with the main focus on positions $X_1$ and $X_2$ located within the core $\Theta$-$X_1$-$X_2$-$\Gamma$ sequence. We identified 13 GABARAP-preferring LIR sequences, and analysed the PLEKHM1-LIR in detail to understand the driving forces of the observed specificity. We propose that residues within the classical LIR sequence, particularly at the $X_1$ and $X_2$ positions, help to define subfamily selectivity and that we can alter selectivity by changing residues in these positions. These data will help define the interaction motifs as either AIM (Atg8), LIR (LC3) or GIM (GABARAP) and develop our understanding of subfamily-specific interactions and their functional consequences.

# Results

### LIR motifs of known autophagy receptors and adaptors feature mATG8 specificity

A high number of autophagy receptors or adaptor structures have been reported, yet the basis for their selective interaction with individual members of the ATG8 family is not well understood. We

speculated whether the LIR motif alone is able to confer selectivity towards a mATG8 subfamily and whether we could derive a subfamily consensus motif from analysis of known mATG8 interaction partners. To address this question, we screened an array of peptides (presented in Fig EV1A and described in Materials and Methods) with the LIR sequences of 30 known and validated autophagy receptors and adaptors (Table 1) against all six human mATG8s for binding (Figs 1A and EV1B and C). In brief, biotinylated peptides were immobilized on streptavidin-coated 96-well plates and incubated with $His_6$-tagged mATG8 proteins. After washing steps, peptide-bound mATG8 was detected in an ELISA reader using anti-His antibodies directly conjugated to HRP (horse radish peroxidase; Fig EV1A).

Due to the wide range of affinities of various LIR sequences towards the LC3/GABARAP proteins, we have normalized our results by dividing values for LC3B interaction by the corresponding value for interaction with GABARAP (Fig 1A, purple bars) and vice

Table 1.  LIR sequences of known mATG8 proteins tested for interactions with all LC3 and GABARAP proteins.

| Protein | Amino acid positions | LIR sequence |
|---|---|---|
| PLEKHM1 | 632–640 | EDE**W**VN**V**QY |
| p62/SQSTM1 | 335–343 | DDD**W**TH**L**SS |
| NBR1 | 729–737 | SED**Y**II**I**LP |
| NDP52 | 130–138 | EED**I**LV**V**TT |
| Tax1BP1 | 137–145 | NSD**M**LV**V**TT |
| OPTN | 175–183 | EDS**F**VE**I**RM |
| NIX/BNIPL3 | 33–41 | NSS**W**VE**L**PM |
| FUNDC1 | 152–160 | DDS**Y**EV**L**DL |
| STBD1 | 200–208 | HEE**W**EM**V**PR |
| c-CBL | 799–807 | SFG**W**LS**L**DG |
| ULK1 | 354–362 | TDD**F**VM**V**PA |
| ULK2 | 350–358 | TDD**F**VL**V**PH |
| ATG13 | 441–449 | HDD**F**VM**I**DF |
| FIP200 | 699–707 | TFD**F**ET**I**PH |
| ATG4B | 4–12 | TLT**Y**DT**L**RF |
| Clathrin HC1 | 511–519 | TPD**W**IF**L**LR |
| Calreticulin | 197–205 | EDD**W**DF**L**PP |
| TP53INP2/DOR | 32–40 | VDG**W**LI**I**DL |
| TP53INP1 | 28–36 | DDE**W**IL**V**DF |
| TBC1D25 | 133–141 | LED**W**DI**I**SP |
| TBC1D5 (LIR1) | 55–63 | RKE**W**EE**L**FV |
| TBC1D5 (LIR2) | 785–793 | DSG**F**TI**V**SP |
| FYCO1 | 1277–1285 | DAV**F**DI**I**TD |
| MAP15K | 337–345 | SRV**Y**QM**I**LE |
| DVL2 | 441–449 | DRM**W**LK**I**TI |
| β-Catenin | 501–509 | PSH**W**PL**I**KA |
| FAM134B | 452–460 | GDD**F**EL**L**DQ |
| KTBD6 | 665–673 | DDF**W**VR**V**AP |
| TECPR2 | 1403–1411 | DLE**D**EW**E**VI |
| JMY | 10–18 | ESD**W**VA**V**RP |

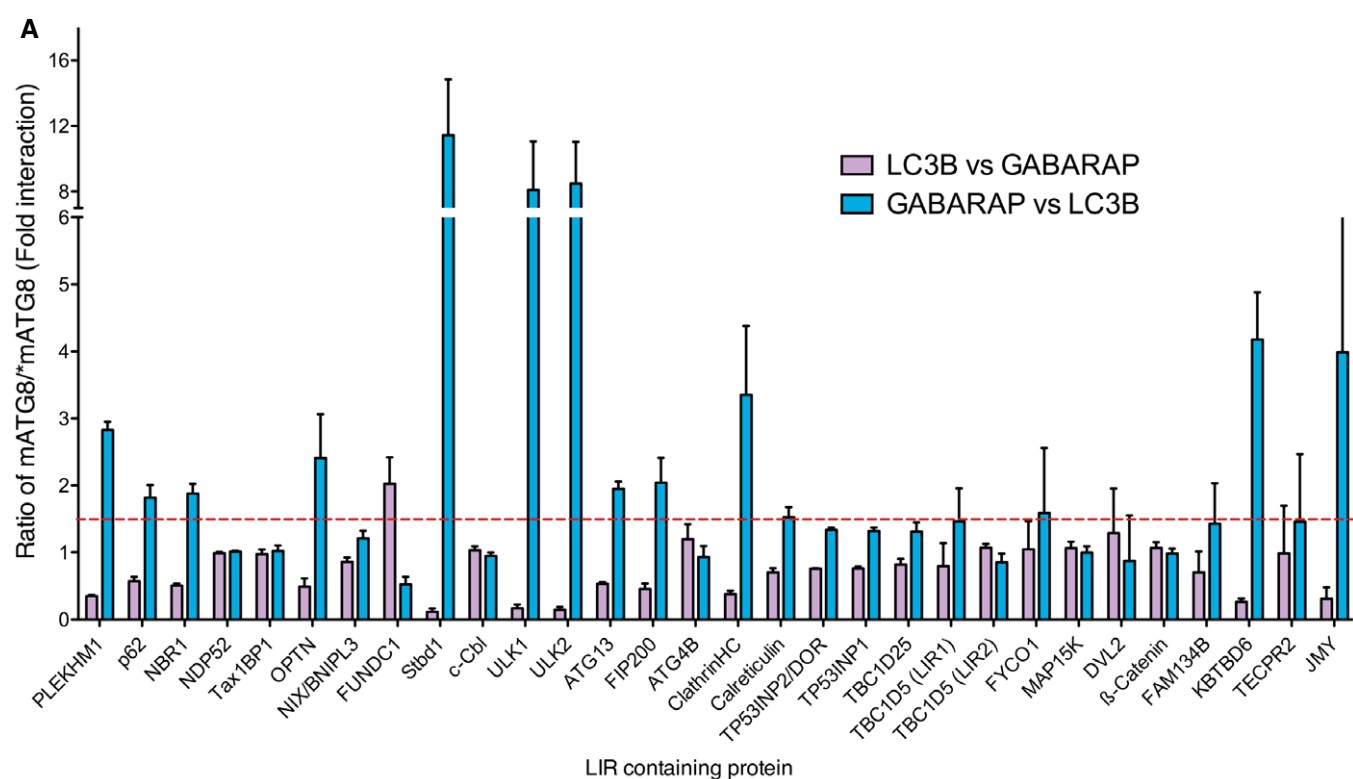

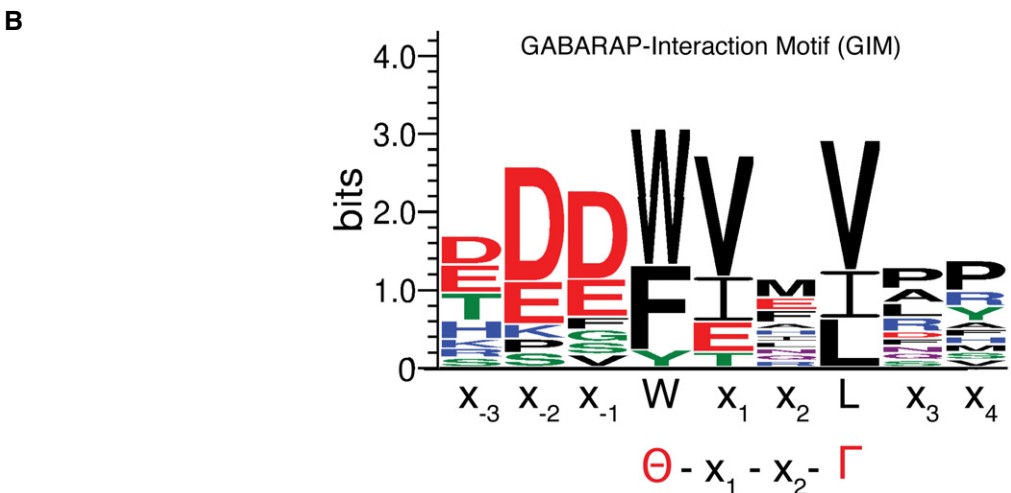

**Figure 1. Defining a mATG8 subfamily-specific interaction motif.**

A   Interaction profile of 30 biotinylated LIR peptides from various proteins against 6xHis-tagged LC3B and 6xHis-tagged GABARAP. Results for each LIR interactions were expressed as either absorbance of LC3B divided by absorbance of GABARAP (purple bars) or absorbance of GABARAP divided by absorbance of LC3B (blue bars) to define whether each LIR shows preference towards either LC3 or GABARAP family proteins. Dashed red line depicts 1.5-fold change cut-off. Values are mean of *n* = 3 independent experiments ± SEM.

B   WebLogo generated from 14 sequences that showed preference towards GABARAP versus LC3B interaction.

versa (Fig 1A, blue bars) to highlight the potential subfamily selectivity of each LIR sequence tested. We classified ratios greater than 1.5-fold as an indication of a preferential interaction towards that particular LC3 or GABARAP family member. Out of the 30 LIRs tested, 12 (40%) showed selectivity towards GABARAP over LC3

subfamily (Table 2) and only one LIR, FUNDC1, preferentially interacted with the LC3 group (Fig 1A). These results are consistent with previously published data, with for example, ULK1/ULK2 and KBTDB6, showing a clear specificity towards GABARAP versus LC3B [20,25]. Using this information, we generated a sequence plot

(Fig 1B) to ascertain whether there were any common sequence features of the GABARAP-specific interaction proteins. In addition to the 12 sequences identified in this experiment as preferential GABARAP subfamily interactors, we also included known GABARAP interactors that were not included in our screen (ALFY and KBTBD7). We found that the fourteen LIR sequences had a high frequency of valine in the $X_1$ position (8 out of 14, 57%) with another three (21%) having an isoleucine (Table 2), indicating that both V and I at position $X_1$ may represent a distinguishing feature of GABARAP-selective LIR sequences. The previously identified PLEKHM1-LIR [19] has a high degree of similarity to this sequence. PLEKHM1 can interact with all LC3s in a GST pull-down assay [19], but we detected a clear preference for binding to GABARAP and GABARAP-L2 over LC3B and LC3C (Figs 1A and EV1B and C). Thus, the isolated LIR of PLEKHM1 shows increased selectivity towards GABARAP family proteins, as opposed to LC3. However, it is unclear whether this is the case *in vivo*.

## PLEKHM1 interacts preferentially with GABARAP family proteins

To further characterize the LIR sequences with preferential binding to GABARAP subfamily proteins, we employed biochemical and biophysical techniques to study interactions of the PLEKHM1-LIR with all six mammalian LC3/GABARAP proteins.

Isothermal titration calorimetry (ITC) experiments titrating purified PLEKHM1-LIR peptide to all six mATG8s (LC3A, LC3B, LC3C, GABARAP, GABARAP-L1 and GABARAP-L2) revealed $K_D$ values in the μM range (Fig 2A and Table 3). Consistent with the previous data (Fig 1A), the GABARAP family proteins had significantly lower $K_D$ values compared to the LC3 family. Indeed, the $K_D$ of GABARAP (0.55 μM) with the PLEKHM1-LIR peptide is approximately eight times lower compared to LC3A (4.22 μM) and approximately 11 times lower compared to LC3B (6.33 μM; Figs 2A and EV2A). In addition, we performed NMR experiments titrating $^{15}$N-labelled LC3A, LC3B, GABARAP-L1 and GABARAP-L2 samples (as representative members of LC3 and GABARAP subfamilies) with the PLEKHM1-LIR peptide. In agreement with the ITC data, we observed

slow exchange behaviour of resonance of the GABARAP subfamily proteins and intermediate exchange for the LC3 subfamily proteins upon titration (Fig EV2B). We mapped the chemical shift perturbations (CSP) of Fig EV2C on the structures of all four proteins used in this experiment (Fig EV2D), revealing a high degree of similarity in the CSP patterns. Most affected are the backbone HN resonances of residues forming the hydrophobic pockets 1 and 2 (HP1 and HP2, highlighted in Fig EV2D), and β-strand two which participates in formation of the intermolecular β-sheet between mATG8 proteins and LIR sequences [20,23,24,30,31].

To probe whether PLEKHM1 has a preference for the GABARAP family *in vivo*, we overexpressed GFP-tagged human ATG8s in the absence and presence of Flag-tagged wild-type PLEKHM1 protein (PLEKHM1-WT-Flag) in HEK293T cells. Immunoprecipitation of GFP-mATG8s revealed that PLEKHM1 strongly co-precipitated with LC3C, GABARAP and GABARAP-L1 (Fig 2B). Previously, we showed that endogenous PLEKHM1 localizes to autolysosomes in the presence of Ku-0063794 (mTOR inhibitor) plus chloroquine (Ku + CQ) to simultaneously increase autophagy flux and block the turnover of autophagosomes [19]. Therefore, we treated HeLa cells overexpressing GFP-mATG8s with Ku + CQ to maximize the capture of endogenous PLEKHM1 interaction with GFP-mATG8s (Fig 2C). Endogenous PLEKHM1 immunoprecipitated preferentially with GFP-GABARAP and GABARAP-L1 (Fig 2C). In contrast, endogenous p62/SQSTM1 co-precipitated with all LC3/GABARAP to a similar extent (Fig 2C). Using either *Plekhm1*$^{+/+}$ or *Plekhm1*$^{-/-}$ (where autophagy is partially blocked) mouse embryonic fibroblasts (MEFs), we were able to show that PLEKHM1 and GABARAP, but not LC3B, formed an endogenous complex when PLEKHM1 was immunoprecipitated after Ku + CQ treatment (Fig 2D) and not under vehicle-only conditions (Fig 2D). This interaction was dependent on PLEKHM1-LIR, as incubation with a PLEKHM1-LIR peptide blocked the interaction but not a scrambled control (Fig 2D). Taken together, these data suggest that PLEKHM1 interacts specifically with GABARAP, but not with LC3B, either *in vitro* or *in vivo*, consistent with the isolated peptide data (Fig 1).

## Understanding the contributing factors to PLEKHM1-LIR specificity towards GABARAPs

To provide a molecular basis for the specificity of the PLEKHM1-LIR interaction with the mATG8 proteins, we solved the crystal structures of PLEKHM1-LIR in complex with the LC3A, LC3C, GABARAP and GABARAP-L1 proteins. In addition, we included in our comparative analysis the structure of the PLEKHM1-LIR:LC3B complex (PDB: 3X0W; McEwan *et al* [19]). Thus, we compared the binding of the same LIR motif across multiple members from both the LC3 and GABARAP subfamilies, an analysis that has not been performed before. To obtain the complex structures, we created chimeric proteins consisting of the mATG8 C-terminally fused to the LIR sequence with a Gly/Ser linker. Crystals diffracted to 2.50 Å for PLEKHM1$^{629–638}$-LC3A$^{2–121}$, 2.00 Å for PLEKHM1$^{629–638}$-GABARAP$^{2–117}$ and 2.90 Å for PLEKHM1$^{629–638}$-GABARAP-L1$^{2–117}$. LC3C could not be crystallized as a chimeric construct, but co-crystals of LC3C with the PLEKHM1-LIR peptide (residues 629–642) diffracted to 2.19 Å resolution. An overview of the structures is provided in Appendix Fig S1 and Appendix Table S1; a detailed analysis of the differences across the LC3/GABARAP proteins in

**Table 2.    LIR sequences of 14 GABARAP-selective interacting proteins.**

| Protein | Amino acid positions | LIR sequence |
|---|---|---|
| PLEKHM1 | 632–640 | EDE<u>W</u>VN<u>V</u>QY |
| ULK1 | 354–362 | TDD<u>F</u>VM<u>V</u>PA |
| ULK2 | 350–358 | TDD<u>F</u>VL<u>V</u>PH |
| KTBD6 | 665–673 | DD<u>F</u>W<u>V</u>RVAP |
| KTBD7 | 665–673 | DEV<u>W</u>VQ<u>V</u>AP |
| JMY | 10–18 | ESD<u>W</u>VA<u>V</u>RP |
| ALFY | 3343–3351 | KDG<u>F</u>IF<u>V</u>NY |
| OPTN | 175–183 | EDS<u>F</u>VE<u>I</u>RM |
| ATG13 | 441–449 | HDD<u>F</u>VM<u>I</u>DF |
| Clathrin HC1 | 511–519 | TPD<u>W</u>IF<u>L</u>LR |
| NBR1 | 729–737 | SED<u>Y</u>II<u>I</u>LP |
| TBC1D5 | 55–63 | RKE<u>W</u>EE<u>L</u>FV |
| STBD1 | 200–208 | HEE<u>W</u>EM<u>V</u>PR |
| p62/SQSTM1 | 335–343 | DDD<u>W</u>TH<u>L</u>SS |

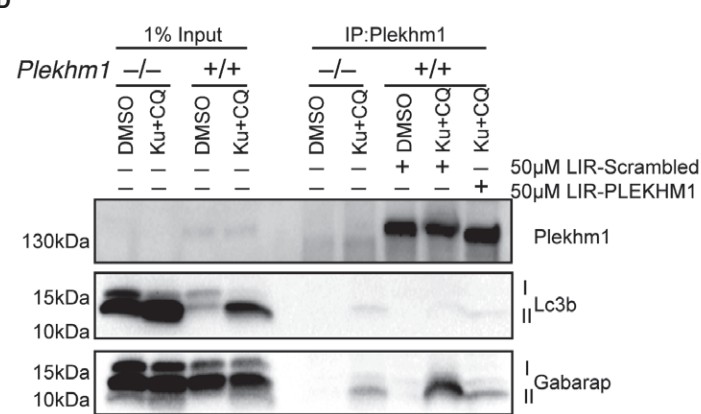

**Figure 2.  PLEKHM1 preferentially interacts with GABARAP *in vitro* and *in vivo*.**

A   ITC titrations of PLEKHM1-LIR peptide into LC3 family proteins (top panel) and GABARAP family proteins (bottom panel). The top diagrams in each ITC plot display the raw measurements, and the bottom diagrams show the integrated heat per titration step. Best fit is presented as a solid line.

B   GFP-tagged LC3/GABARAP proteins were expressed alone or with PLEKHM1-WT-Flag in HEK293T cells and immunoprecipitated using GFP-Trap beads and blotted for the presence or absence of PLEKHM1 (anti-Flag tag). Free GFP was observed in lanes three to six (GFP-LC3A and GFP-LC3B) potentially due to lysosomal turnover.

C   GFP-LC3/GABARAPs were overexpressed in HeLa cells and treated for 4 h with KU-0063794 (10 μM) plus chloroquine (20 μM), immunoprecipitated with GFP-Trap beads and blotted for the presence of endogenous PLEKHM1.

D   *Plekhm1*[+/+] *or Plekhm1*[−/−] mouse embryonic fibroblasts were either treated with vehicle (DMSO) or treated for 4 h with KU-0063794 (10 μM) plus chloroquine (20 μM). Samples were lysed in NP-40 lysis buffer, and endogenous PLEKHM1 was immunoprecipitated in the presence of 50 μM PLEKHM1-LIR peptide (KVRPQQ**EDEWVNV**QYPDQPE) or 50 μM Scrambled (Scr) PLEKHM1-LIR peptide (VQEQQEPPPVKNYDVEQWDR). Samples were then immunoblotted for the presence of endogenous PLEKHM1, LC3B and GABARAP proteins.

Source data are available online for this figure.

**Table 3. Thermodynamic parameters of interactions between LC3/GABARAP proteins and PLEKHM1-LIR peptide.**

| | $\Delta H$ (kcal mol$^{-1}$) | $\Delta S$ (cal mol$^{-1}$ K$^{-1}$) | $-T \times \Delta S$ (kcal mol$^{-1}$) | $\Delta G$ (kcal mol$^{-1}$) | $K_A$ (×10$^6$ M$^{-1}$) | $K_D$ (μM) | $N$ |
|---|---|---|---|---|---|---|---|
| PLEKHM1-LIR WT | | | | | | | |
| LC3A | −7 ± 0.2 | +1.17 | −0.35 | −7.33 | 0.24 ± 0.02 | 4.22 | 0.98 ± 0.02 |
| LC3B | −5.8 ± 0.2 | +4.27 | −1.27 | −7.09 | 0.16 ± 0.01 | 6.33 | 1.06 ± 0.01 |
| LC3C | −8.3 ± 0.2 | −2.83 | +0.84 | −7.48 | 0.29 ± 0.02 | 3.45 | 0.99 ± 0.02 |
| G$_{ABARAP}$ | −10.6 ± 0.1 | −6.92 | +2.06 | −8.54 | 1.8 ± 0.1 | 0.55 | 1.00 ± 0.01 |
| G$_{ABARAP}$-L1 | −7.8 ± 0.1 | +1.94 | −0.58 | −8.35 | 1.3 ± 0.1 | 0.77 | 1.00 ± 0.01 |
| G$_{ABARAP}$-L2 | −6.1 ± 0.1 | +7.00 | −2.09 | −8.23 | 1.07 ± 0.03 | 0.93 | 1.05 ± 0.01 |
| PLEKHM1-LIR W635A | | | | | | | |
| LC3B | −0.8 ± 0.1 | +15.4 | −4.59 | −5.44 | 0.01 ± 0.01 | > 100 | 1* |
| G$_{ABARAP}$ | −1.1 ± 0.1 | +14.4 | −4.29 | −5.40 | 0.01 ± 0.01 | > 100 | 1* |
| PLEKHM1-LIR V636G | | | | | | | |
| LC3B | −1.7 ± 0.1 | +12.4 | −3.70 | −5.42 | 0.01 ± 0.01 | > 100 | 1* |
| G$_{ABARAP}$ | −7.5 ± 0.1 | −5.66 | +1.69 | −5.83 | 0.02 ± 0.00 | 52 | 1* |
| PLEKHM1-LIR N637G | | | | | | | |
| LC3B | −1.4 ± 0.1 | +15.0 | −4.47 | −5.87 | 0.02 ± 0.01 | 50 | 1* |
| G$_{ABARAP}$ | −6.7 ± 0.1 | +1.07 | −0.32 | −6.97 | 0.13 ± 0.01 | 7.75 | 1.17 ± 0.04 |
| PLEKHM1-LIR VNV-CIL | | | | | | | |
| LC3B | −7.9 ± 0.1 | +3.30 | −0.98 | −8.84 | 3.03 ± 0.27 | 0.33 | 1.06 ± 0.01 |
| G$_{ABARAP}$ | −9.2 ± 0.1 | +1.49 | −0.44 | −9.62 | 11.3 ± 0.38 | 0.09 | 1.07 ± 0.01 |

*For the weak interactions, the number of binding sites $N$ was fixed to 1 upon fitting.

complexes with PLEKHM1 LIR peptides is also provided in the Appendix. The relevant findings are summarized below.

We compared the LIR-bound and LIR-unbound GABARAP family structures to the LC3 family structures to assess whether global conformational changes account for the preference of PLEKHM1-LIR towards GABARAP. The structures of the PLEKHM1-LIR-bound mATG8 proteins overlay very closely (Fig EV3A), and exhibit conventional pattern of LIR:mATG8 interactions, although subtle differences were observed (Fig EV3C–G and Appendix Table S2).

Next, we analysed the microenvironment surrounding the four key PLEKHM1-LIR residues W635, V636, N637 and V638 consisting of the core $\Theta$-X$_1$-X$_2$-$\Gamma$ motif when bound to mATG8 proteins. The HP1 and HP2 pockets are known to be critical for the LIR interaction, and the tighter packing of the two essential residues W635 and V638 into HP1 ($\Theta$) and HP2 ($\Gamma$) of GABARAPs versus LC3 families (Fig 3A and D; results in Appendix Table S3) may in part explain the generally stronger binding of PLEKHM1-LIR to GABARAP proteins.

Our structural analysis revealed that PLEKHM1-LIR residues in positions of X$_1$ and X$_2$ also participate in the binding and could be important for the subfamily-specific interaction's network (Figs 3B and C, and EV4 and results in Appendix). Residue N637 at the X$_2$ position formed more preferential contacts for binding of the GABARAP proteins due to better geometry of an intermolecular hydrogen bond to an invariant arginine residue in all GABARAP proteins that is lysine in all LC3 proteins (Figs 3C and EV4B, and Appendix Table S3). In contrast, for the V636 in the X$_1$ position, we did not observe significant differences in the intermolecular contacts (Fig 3B). However, we observed that in all LC3 subfamily proteins,

the surface to which V636 binds was stabilized by an intramolecular salt bridge, which is absent in GABARAP subfamily structure (Fig EV4A).

Taken together, our structural analysis reveals that residues in PLEKHM1-LIR positions $\Theta$, $\Gamma$ and X$_2$ form GABARAP subfamily-favourable contacts, while V636 in the X$_1$ position has LC3 subfamily-favoured contacts.

### X$_1$ and X$_2$ residues are important for PLEKHM1-LIR:mATG8 interaction

To find contributing factors of the interactions and to analyse in greater detail how selectivity could be achieved, we complemented our structural studies with peptide arrays of the PLEKHM1-LIR by mutating each position to alanine. PLEKHM1-LIR WT peptide (EDEWVNVQY) reproducibly reflected the ITC data (Fig 2A and Table 3) where PLEKHM1-LIR WT with GABARAP (green bar) shows the most potent interaction, followed by GABARAP-L1, -L2, LC3C and LC3A, with LC3B as the weakest interactor (Fig 3E). W635A was sufficient to abolish all PLEKHM1-LIR:mATG8 interactions (Fig 3E), V638A abolished LIR-LC3 family as well as LIR–GABARAP-L1 interactions, but only reduced GABARAP and GABARAP-L2 interactions (Fig 3E), and W635A/V638A completely disrupted all LIR–mATG8 interactions (Fig 3E). Therefore, we are confident that our experimental set-up can be used to accurately assess any alterations in LIR:mATG8 interactions introduced by mutation.

Through substitution of W635 and V638 for residues found in other LIR sequences, we showed that W635F and W635Y mutants

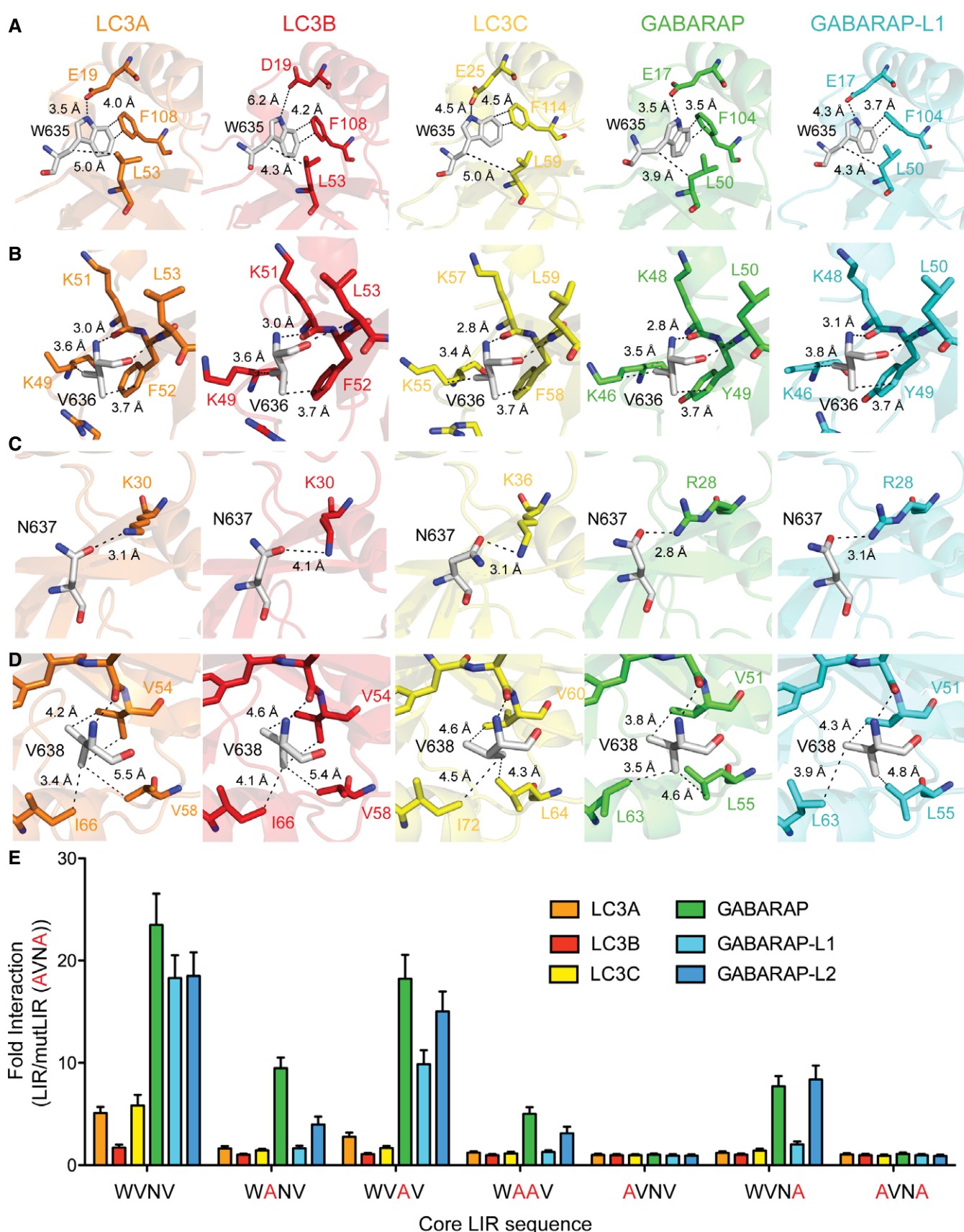

**Figure 3.**

Figure 3.   **Importance of residues in PLEKHM1-LIR for preferential binding of GABARAP subfamily proteins.**

A   Sections of the complex structures representing W635 of PLEKHM1 and its microenvironment. Network of intermolecular contacts for the PLEKHM1-LIR residue W635 within complexes with each mATG8 proteins (indicated on each plot). Partner residues in each mATG8 protein are given W635 of PLEKHM1 interacts with E19 of LC3A with 3.5 Å distance, with D19 of LC3B and E25 in LC3C in similar way, but the bond distances are higher (6.2 Å and 4.5 Å), suggesting a weaker interaction. W635 interacts with E17 in GABARAP and GABARAP-L1 with distances of 3.5 Å and 4.3 Å. Additionally, aromatic carbons of W635 are significantly closer to the carbons of GABARAP non-polar residues, forming the HP1 (Appendix Table S3).

B   Sections of complex structures representing V636 of PLEKHM1 and its microenvironments. V636 in the $X_1$ position of PLEKHM1 interacts with residues at the surface of the mATG8 protein. This includes hydrophobic interactions with the aromatic residue (phenylalanine in LC3 and tyrosine in GABARAP) and lysine for both families, and for the LC3 protein, an arginine also forms part of the interaction surface with V636. In contrast, this arginine in the GABARAP family of proteins is further away and more disordered.

C   Sections of complex structures representing N637 of PLEKHM1 and its microenvironments. For the LC3 subfamily proteins, the hydrogen bonding distance of N637 (LIR) with K30 (LC3) correlates with binding affinity to the LIR peptide. The bond distances are (average for all monomers in ASU): 4.1 Å for LC3B, $K_D$ = 6.3 μM; 3.1 Å for LC3A, $K_D$ = 4.2 μM; and 3.1 Å for LC3C, $K_D$ = 3.5 μM. For LC3A, two of the four monomers in the ASU do not show this interaction and in LC3C, one of the eight monomers in the ASU do not have this interaction, suggesting that this interaction is variable in the LC3 structures. In comparison, R28 in the GABARAP subfamily proteins is always hydrogen-bonded to N637 with a generally shorter bond distance and the geometry of the hydrogen bond between the arginine and asparagine is close to optimal for a hydrogen bond (N-H···O angles are as follows: LC3A 16.6°, LC3B 48.3°, LC3C 26.7°; GABARAP 4.7°, GABARAP-L1 6.9°). The average hydrogen bond distance for all monomers in ASU is as follows: 2.8 Å for GABARAP, $K_D$ = 0.6 μM; and 3.1 Å for GABARAP-L1, $K_D$ = 0.8 μM.

D   Sections of complex structures representing V638 of PLEKHM1 and its microenvironments. Tighter packing of V638 in HP2 of GABARAP subfamily proteins is observed. V51 GABARAPs' side chains are in close proximity to the PLEKHM1 V638 (3.8 and 4.3 Å for GABARAP and GABARAP-L1, respectively), while side chains of residues in equivalent positions of LC3 subfamily proteins are further away (LC3A/B/C V54/V54/V60 − 4.2/4.6/4.6 Å). Similarly, GABARAPs' L55 side chain are closer to the PLEKHM1 V638 (4.6 and 4.8 Å for GABARAP and GABARAP-L1, respectively); LC3A/LC3B/LC3C V58/V58/L64 are distanced to PLEKHM1 V638 at 5.5/5.4/4.3 Å. Additionally, the V638 side chain shows some rotational flexibility, observed for when comparing all the crystal structures.

E   Biotinylated PLEKHM1-LIR peptides (WT and alanine substitutions of highlighted residues) were incubated with streptavidin-coated plates, washed and subsequently incubated with 6xHis-tagged mATG8 proteins (human LC3A, -B, -C, GABARAP, -L1 and -L2 proteins). These were washed and incubated with anti-His-HRP to detect His-tagged mATG8s directly bound to biotinylated PLEKHM1-LIR peptides. Samples were again washed and incubated with TMB substrate (3,3′,5,5′-tetramethylbenzidine). After 5 min of incubation time, the reaction was stopped by addition of acid and the sample absorption was directly read at 450 nm. Results were normalized to absorbance of the PLEKHM1-mutLIR (EDE**A**VN**A**QY) where both hydrophobic core residues were substituted with alanine and expressed as a fold change of mutant LIR (background noise). Results shown are mean ± SD of *n* = 4 independent experiments.

weakened the interaction with all six mATG8s (Fig EV5A). Interestingly, V638L or V638I substitutions did not affect the interactions of PLEKHM1-LIR to the GABARAP family or LC3A and LC3C proteins, but did increase the affinity of the interaction in LC3B (Fig EV5A). Overall, W635 and V638 act as the corner stones for LIR–mATG8 interaction, where the large aromatic W side chain is optimal for all mATG8s, but where the HP2 pocket that binds V638 is able to accommodate slightly larger (extra methyl) I or L residues, perhaps due to the additional conformational flexibility of the I and L side chains compared to V.

Next, we assessed the effect of an alanine mutation of the $X_1$ and $X_2$ residues, V636 and N637, respectively, on the interactions of PLEKHM1-LIR with mATG8s. Surprisingly, the V636A substitution had a similar effect as V638A and abolished the interaction of PLEKHM1-LIR with all LC3 and GABARAP-L1 but only reduced GABARAP and GABARAP-L2 interactions (Fig 3E). On the other hand, N637A mutation had only a mild effect on the interaction with the GABARAP family but strongly reduced LC3A, LC3B and LC3C interactions (Fig 3E).

Taken together, our data indicate that residues in PLEKHM1-LIR positions $X_1$ and $X_2$ may provide a means of fine-tuning the selectivity of LIRs towards LC3 or GABARAP subfamilies.

### Residues at positions $X_1$ and $X_2$ provide refinement of selective LIR–mATG8 interactions

To study the role of the amino acids in positions $X_1$ and $X_2$ in more depth, we substituted V636 (Fig EV5C) and N637 (Fig EV5D) of PLEKHM1-LIR with all other 19 amino acids and analysed the relative affinity of each mutated peptide to all six mATG8 proteins in our peptide array (normalizing the strength of interaction in each individual case to that for PLEKHM1-LIR WT). We included the

W635A/V638A double mutant (PLEKHM1-mutLIR) as a negative control (Fig 3E). This allowed us to assess mutations that either increased or decreased the interaction with each mATG8 subfamily member, relative to the PLEKHM1-LIR WT sequence. Firstly, we found that substitution of V636 had for most residue types a negative influence on both LC3 and GABARAP (Fig EV5C) family interactions, particularly when mutated to G, K, R, P or S, indicating that the amino acid in position $X_1$ can have a profound impact on LIR-mATG8 interactions. For V636G, we confirmed these data by ITC (Fig 4A). Notably, V636C was the only mutant that increased its interaction with any mATG8, specifically LC3B, but did not affect overall interactions with LC3A, LC3C or GABARAP family members (Fig EV5C). Next, we tested the effect of mutating N637 ($X_2$) of PLEKHM1-LIR. Overall, substitution of N637 to G or P completely disrupted LIR-LC3 family interactions, with only a mild effect on all GABARAPs (Figs 4A and EV5D). We also found that mutation of N637 to either C, F, I, L, V, W or Y enhanced the interaction of PLEKHM1-LIR with LC3B up to fivefold compared to WT PLEKHM1-LIR but only mildly affected LC3A, LC3C or GABARAP family members (Fig EV5D).

Using combinations of amino acid that individually increased PLEKHM1-LIR:LC3B interaction, we could show that mutation of the core WVNV motif to either WCIL, WCFL or WCVL increased the interaction of PLEKHM1-LIR with LC3B (Fig 4B). Indeed, using a WCIL core sequence resulted in a fivefold increase in GABARAP interaction but a greater than 20-fold increase in the LC3B interaction ($K_D$ 0.3 μM; Fig 4A and B). This was mirrored *in vivo* with the PLEKHM1-WCIL (full length) showing increased co-precipitation with GFP-LC3B from cell lysates compared to PLEKHM1-LIR WT and PLEKHM1-mutLIR (Fig 4C). Thus, we were able to show that specific alterations in the LIR motif of full-length PLEKHM1 can shift its selectivity towards LC3B.

**Autophagy adaptors and receptor proteins with altered mATG8 subfamily selectivity**

Finally, we wanted to apply and subsequently verify our findings by targeted mutation of established autophagy players: p62, FUNDC1 and FIP200. Due to their impact, we specifically substituted existing residues at positions $X_1$ and $\Gamma$ of LIRs either singly or in combination to valine, thus driving the LIR sequences towards our GIM consensus sequence (Fig 1). Firstly, using the established autophagy receptor protein p62/SQSTM1 as a model LIR (DDD**W**TH**L**SS) that interacts strongly with LC3B ($K_D \sim 1.5$ μM), we tested whether substitution of T339V ($X_1$) and L341V ($\Gamma$) altered the selectivity of p62/SQSTM1 LIR *in vivo*. We immunoprecipitated GFP-mCherry-tagged wild-type and mutant forms of p62/SQSTM1 from HEK293 cells. Under basal conditions, p62/SQSTM1-WT co-precipitated with endogenous LC3B and weakly with endogenous GABARAP (Fig 5A). The p62/SQSTM1 T339V mutant presented a striking shift in the interaction with endogenous GABARAP over LC3B (Fig 5A) while L341V alone having a mild effect (Fig 5A). However, a double T339V/L341V substitution showed a strongly enhanced shift towards GABARAP with only a moderate increase in endogenous LC3B interaction (Fig 5A). Similarly, we were able to enhance the interaction between FIP200 (LIR sequence FD**FETI**PH) and GABARAP in this instance by the introduction of two V residues into the $X_1$ and $\Gamma$ sites (**FVTV**; Fig 5B). Lastly, we tested the only LC3-selective LIR in our peptide array present in the mitochondrial autophagy receptor FUNDC1 (DDS**YEVL**DL; Fig 1A). Similar to p62/SQSTM1, substitution of E19V moderately enhanced the interaction with GABARAP but, in this instance, also increased LC3 interaction (Fig 5C, left panels), whereas L21V alone had little effect on both LC3 and GABARAP interaction. However, strikingly, the double-substitution E19V/L21V (Y**V**V**V**) enhanced the interaction with GABARAP under non-stimulated conditions (Fig 5C, left panel), which was further enhanced in the presence of the mitochondrial decoupling agent CCCP (Fig 5C, right panels).

Overall, we demonstrate that LIR residues at the $X_1$ and $\Gamma$ positions are important for defining GABARAP-selective LIR sequences (GABARAP Interaction Motif: GIM) that are found in a number of endogenous proteins. Moreover, we can alter the selectivity of known autophagy adaptors and receptors by introducing valine residues in the $X_1$ and $\Gamma$ positions to enhance the interaction with GABARAPs, or by mutating $X_2$ and $\Gamma$ positions to enhance the interaction with LC3s. In conclusion, the previously unassigned $X_1$ and $X_2$ positions of a classical $\Theta$-$X_1$-$X_2$-$\Gamma$ sequence are important regulators of LC3 and GABARAP subfamily selectivity of LIRs and help define a GABARAP subfamily selective interaction motif, namely W-[V/I]-$x_2$-V.

# Discussion

The process of building, shaping and "filling" an autophagosome requires a large number of proteins with distinct functions. These include E1-, E2- and E3-like enzymes, kinases, scaffold and adaptor proteins that help build and transport autophagosomes to their destination. At the core of this process are the small ubiquitin-like modifiers, the ATG8-like proteins, that are conjugated onto the growing autophagosome on both the convex and concave surfaces of the nascent autophagosome. The critical positioning of these proteins

allows them to recruit both adaptors (present on convex side and that are not degraded in an autophagy-dependent manner) and receptors (present on concave side that are degraded along with the cargo) to the autophagosome [32]. In all cases, the interaction with mATG8 proteins is mediated through a direct interaction between a LIR/AIM motif on the receptor/adaptor and two hydrophobic pockets on the ATG8 proteins. This interaction was first described for the prototypical autophagy receptor protein, p62/SQSTM1, which linked autophagy-mediated protein aggregate degradation with LC3B conjugation on the autophagosome [7]. Since then, there has been a deluge of both adaptors and receptors identified with conserved LIR motifs that conform to the $\Theta$-$X_1$-$X_2$-$\Gamma$ motif. These include autophagy adaptor proteins such as PLEKHM1, ULK1/2, TBC1D5, KBTBD6/7, ALFY and JMY and link the autophagosome to various cellular machineries, such as the autophagosome initiation complex and autophagosome–lysosome fusion machinery [18–20,25,26,33]. Autophagy receptors on the other hand include FAM134B, OPTN, TAX1BP1, NDP52 and p62/SQSTM1 and are linked to the direct removal of a variety of cellular structures including pathogens, protein aggregates, peroxisomes, mitochondria, ER turnover and removal of ferritin aggregates (reviewed in [3]).

However, despite the ever-increasing number of LC3/GABARAP interaction partners identified and perhaps the over-reliance on LC3B as the main marker of autophagosomes, there is now emerging distinct roles of each LC3 and GABARAP subfamily. For example, both LC3 and GABARAP families are essential for autophagy flux [29]; however, LC3s were reported to be involved in phagophore extension and GABARAPs required for autophagosome closure [29]. Moreover, GABARAP can activate ULK1 complex to initiate autophagy, irrespective of its conjugation status [28]. Indeed, this is also reflected in *C. elegans* homologues of GABARAP (LGG-1) and LC3 (LGG-2), where LGG-1 interacts with the Unc51/EPG-1 (ULK1/ATG13) and LGG-2 binds to LGG-3 and ATG-16 [34]. Overall, there appears to be an evolutionary separation of function of LC3s versus GABARAPs where there may be a preference for GABARAPs conjugated to PE on the convex autophagosomal surface to engage adaptors, and LC3s on the concave side to recruit receptors and cargo. However, there are some interesting exceptions. For example, OPTN-LIR in its unmodified state clearly shows preference for GABARAP, however when activated through TBK1 phosphorylation at S177, switches to LC3B indicating a functional shift between GABARAP and LC3 families [10,31]. Also, FYCO1 (LC3A-specific adaptor) and NBR1 (GABARAP-L1-specific receptor) are other exceptions that require further exploration [27,30]. Since the initial identification and characterization of the p62/SQSTM1 LIR, there has been little headway in the identification of LC3 or GABARAP subfamily-selective LIR sequences. Currently, there is only one subfamily-specific LIR sequence, CLIR, present in NDP52 and TAX1BP1 [11,21] that specifically mediates the interaction with LC3C.

For the first time, we provide evidence of a GABARAP-selective LIR motif built around the classical $\Theta$-$X_1$-$X_2$-$\Gamma$ motif and indicate derivations that support LC3B binding. Using a peptide-based array to test interaction profiles of known LIRs, we found that 14 out of 30 tested had a strong preference for GABARAP versus LC3B. These included ULK1/2 and KBTBD6, which had previously been shown to be GABARAP specific [20,25], and several that previously had not been identified as GABARAP selective, including JMY and PLEKHM1. Interestingly, PLEKHM1 showed a strong preference for

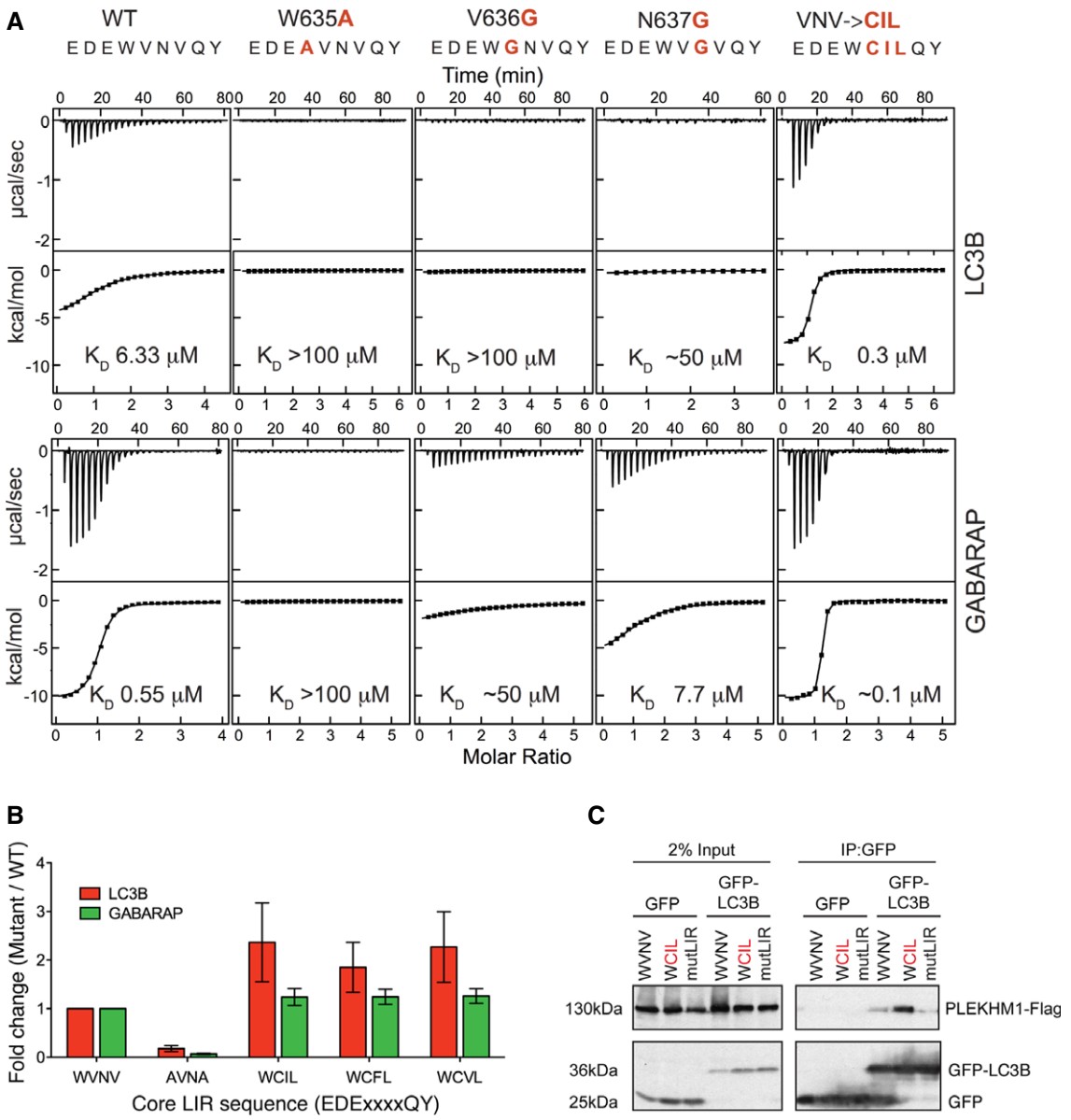

**Figure 4. Mutation of $X_1$ and $X_2$ positions differentially affect interaction with LC3 and GABARAP proteins.**

A   ITC titrations of mutated PLEKHM1-LIR peptide into LC3B (top panel) and GABARAP (bottom panel) proteins. The top diagrams in each ITC plot display the raw measurements, and the bottom diagrams show the integrated heat per titration step. Best fit is presented as a solid line. Mutations within the PLEKHM1-LIR peptide are indicated at the top of the figure.

B   Biotinylated peptides of PLEKHM1-LIR WT (EDE**WVNV**QY), PLEKHM1-mutLIR (EDE**A**VN**A**QY) or mutants that increase LC3B interaction (EDE**WCIL**QY; EDE**WCFL**QY; EDE**WCVL**QY). Results shown are mean ± SEM of *n* = 5 independent experiments.

C   Co-immunoprecipitation of GFP alone or GFP-LC3B with PLEKHM1-WT-Flag, mutant LIR (mutLIR; EDEWVNV/AAAAVNG) or variant LIR (WVNV/WCIL). Free GFP was observed after co-expression of PLEKHM1-WT with GFP-LC3B and not LIR mutants of PLEKHM1 potentially due to lysosomal turnover.

Source data are available online for this figure.

GABARAP versus LC3 despite the apparent similarity of PLEKHM1-LIR (EDEWVNV) with p62/SQSTM1 LIR (DDDWTHL). Indeed, while this manuscript was under review, it was shown that in cells that lacked all GABARAP family members, PLEKHM1 failed to localize to vesicles surrounding damaged mitochondria [35]. The majority of proteins we identified as more selective towards GABARAP presented with a valine/isoleucine in the $X_1$ position and a valine/isoleucine in the Γ position (64%). Indeed, using mutational analysis

of the $X_1$ and $X_2$ positions of PLEKHM1-LIR, which have previously not been linked to LC3 or GABARAP subfamily interactions, we were able to show that residue $X_1$ is important for the LC3 and GABARAP interface. For example, substitution of V636 with small G, A, P, S or positively charged N, K, R and H residues are generally disruptive to LIR-mATG8 interactions. The effect of these substitutions is mediated by specific side chain structure, orientation and mobility, and not by the ability of mutated PLEKHM1-LIR to adopt a

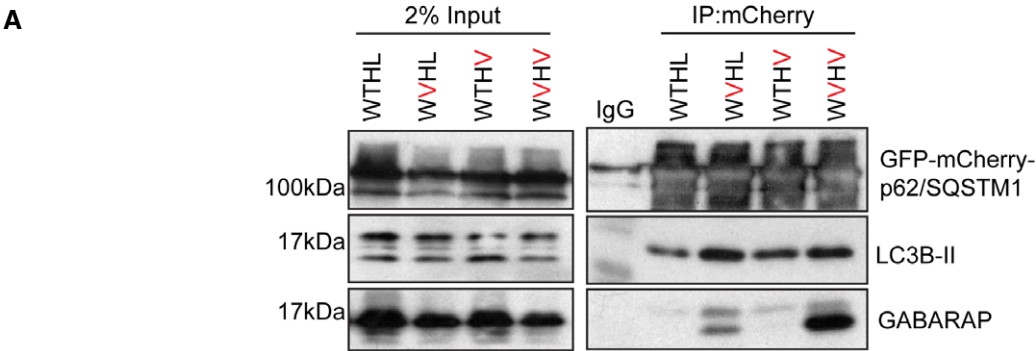

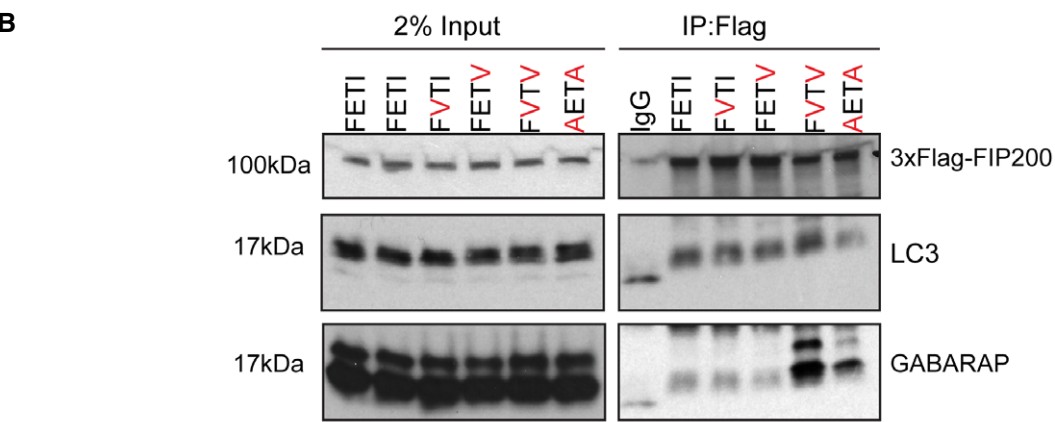

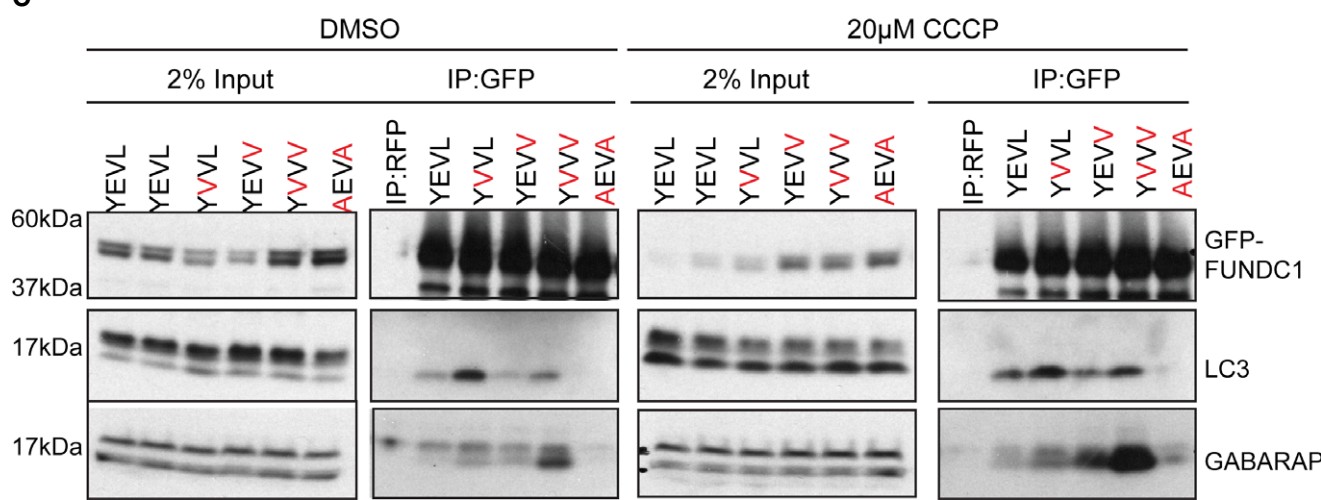

**Figure 5.  Autophagy adaptor and receptor proteins with altered mATG8 subfamily selectivity.**

A   GFP-mCherry-p62/SQSTM1 WT, T339V, L341V and T339V/L341V were overexpressed in HEK293 cells and immunoprecipitated using anti-RFP beads and subjected to SDS–PAGE and Western blotting. Blots were probed for the presence of endogenous GABARAP and LC3B proteins.

B   3xFlag-FIP200 WT, E703V, I705V, E703V/I705V and F702A/I705A were overexpressed in HEK293 cells and immunoprecipitated using anti-Flag beads and subjected to SDS–PAGE and Western blotting. Blots were probed for the presence of endogenous GABARAP and LC3B proteins.

C   GFP-FUNDC1 WT, E19V, L21V, E19V/L21V and Y18A/L21A were overexpressed in HEK293 cells and either treated with vehicle only (DMSO) or 20 μM CCCP for 2 h, lysed and GFP-FUNDC1 immunoprecipitated using anti-GFP beads (or anti-RFP beads as control) and subjected to SDS–PAGE and Western blotting. Blots were probed for the presence of endogenous GABARAP and LC3B proteins.

Source data are available online for this figure.

β-stranded conformation (see results in Appendix for details). We found a more favourable microenvironment of PLEKHM1-LIR $X_1$ (V636) for LC3 subfamily structures than for GABARAP subfamily structures (Figs 3B and EV4A and results in Appendix). However, we believe that the observed differences do not provide enough energy to shift the preference of PLEKHM1-LIR towards LC3 proteins and are not reproducible for other LIR:GABARAP structures. For example, the structure of KBTBD6-LIR with core sequence W-V-R-V in complex with GABARAP [19] displays similar microenvironment features of V at $X_1$ position as PLEKHM1-LIR V636 in complexes with LC3 proteins. Therefore, the microenvironments of V636 are similar in all PLEKHM1-LIR:mATG8 complexes and, when mutated, results in a universal decrease in interaction with all mATG8s (Fig EV5C). We also show that substitutions at position $X_2$ (N637) are less disruptive; however G and P can also decrease most LIR–mATG8 interactions *in vitro*. When we introduce either K or R in the $X_2$ position of PLEKHM1-LIR, thereby making it similar to KBTBD6-LIR (DDFWV**R**VAP) that forms an intermolecular hydrogen bond with GABARAP Y25 [20], we observed a reduced interaction with GABARAPs indicating that although similar in sequence, other factors, such as the F in the $X_{-1}$ position, may also influence selectivity. Perhaps the most surprising results were when we mutated $X_2$ (N637) to C, F, I, L, V, W or Y, resulting in a large increase in the interaction with LC3B only, compared to WT PLEKHM1-LIR peptide. Indeed, when we rationally mutate the $X_1$, $X_2$ and the $\Gamma$ positions of PLEKHM1-LIR using combinations that increase LC3B interaction, we can achieve a direct 20-fold increase in the interaction with LC3B using ITC as a measurement.

This alteration is not confined to PLEKHM1, as we show that by introducing a single point mutation in the $X_1$ position of p62/SQSTM1-LIR, T339V, we can increase the interaction of p62/SQSTM1 with endogenous GABARAP. Interestingly, the p62/SQSTM LIR shows slight preference for GABARAP in isolated LIR-peptide assays. However, when immunoprecipitated, p62/QSQTM1 clearly shows a preference for LC3 interaction (Fig 5A). It is unclear why this may be the case, but could be due, in part, to its ability to dimerize through its PB1 domain, resulting in a conformation that is preferential for LC3 over GABARAP in cells. We tested the effect of substitution of a recently identified ALS-FTD p62/SQSTM1 mutation ($\Gamma$ position, L341V) that has been associated with poor prognosis [36]. We showed that the L341V mutation alone had little effect on LC3/GABARAP-specific interaction. However, when we combine T339V and L341V (T329V/L341V), the interaction is dramatically switched towards endogenous GABARAP with little or no effect on the interaction with LC3B interaction. Interestingly, while this manuscript was in preparation, the only LC3B-specific LIR identified in our peptide screen, FUNDC1 (Fig 1A), was shown to have specificity for, and a non-canonical mode of interaction with, LC3B [37], where position $X_2$ (V20) is inserted alongside Y18 into HP1 of LC3B [37]. This may provide a structural explanation for our own data, where mutation of Plekhm1 N637 ($X_2$) to V (or I) results in enhanced interaction with LC3B (Fig EV5D). Upon identification of additional LC3B-specific interactors, the inclusion of V/I in position $X_2$ may turn out to be critical for LC3B specificity. In addition to p62/SQSTM1, we were able, through mutagenesis of positions $X_1$ and $\Gamma$ to V, to enhance the interaction of both FUNDC1 and FIP200 with endogenous GABARAP over LC3B indicating a more general consensus sequence for GABARAPs.

This leads us to propose for the first time a subfamily-selective LIR sequence that we have termed GABARAP Interaction Motif (GIM; [W/F]-[V/I]-$X_2$-V). Despite extensive efforts, we were unable to identify a similar set of LIRs with clear preference for the LC3 subfamily (specifically LC3B). Analysed LIRs that did not show a clear GABARAP preference showed rather equal binding to LC3B and GABARAP (Fig 1A). This indicates that *in vivo* LC3B preference might not be defined by a LIR motif with lacking GABARAP affinity but rather by a LIR motif with an LC3B affinity that is in the same range as its GABARAP affinity (Fig 4). Additional domains, as for example the dimerization domain of p62 or post-translational modifications (phosphorylation) might in those cases tip the scales towards a clear preference for LC3B *in vivo*. The identification of a GIM (and its separation from the LIR) will allow more precise and directed autophagy research towards understanding adaptor- and/or receptor-specific function within the life cycle of an autophagosome and the role of mammalian ATG8 paralogues during autophagosome formation, cargo selection, transport and fusion.

## Materials and Methods

### Cloning plasmid preparation

The genes for the truncated LC3A$^{2–121}$, LC3C$^{8–125}$, GABARAP$^{2–117}$ and GABARAP-L1$^{2–117}$ proteins were cloned into pET30ΔSE vector between the *Bam*HI and *Xho*I sites using previously established protocols [38]. The chimeric constructs of the PLEKHM1-LIR attached to the LC3A, GABARAP and GABARAP-L1 proteins were prepared by inserting the oligonucleotide sequence corresponding to the PLEKHM1-LIR peptide (P$^{629}$QQEDEWVNV$^{638}$) and glycine–serine linker into the *Bam*HI site of the pET30ΔSE vector, placing the PLEKHM1-LIR at the N-terminal of the mature chimeric protein (similar to [31,38]). For the expression of human LC3 and GABARAP proteins for ITC and NMR experiments, plasmids with appropriate modified Ub-leaders in pET vectors were used [39]. Gene, encoding PLEKHM1-LIR peptide, was ordered as synthetic oligonucleotides (Eurofins Genomics GmbH) and cloned into the pET39_Ub63_ vector [39] by NcoI–BamHI restriction sites. After TEV cleavage, the resulting peptide has the amino acid sequence GAMG-P$^{629}$QQEDEWVNVQYPD$^{642}$, where the first four residues (GAMG) are the cloning artefact.

### Protein expression and purification

The chimeric constructs were expressed as a His-tag fusion protein in *E. coli* BL21(DE3) cells. The cells were induced with 0.3 mM IPTG at OD$_{600}$ 0.6 for 16 h at 26°C. The cell pellets were lysed using mechanical sonication in lysis buffer (20 mM Tris–HCl pH 9.0, 100 mM NaCl, 10 mM imidazole, supplemented with 0.1% Triton X-100). The proteins were purified using Ni-NTA beads (GE Healthcare) and the His-tag was cleaved using thrombin (Invitrogen) at room temperature for 16 h. The last step was gel filtration chromatography using a Superdex S200/300 GL column (GE Healthcare). The proteins were concentrated using spin concentrators (Vivaspin). For ITC and NMR studies, the non-labelled and stable isotopes labelled LC3 and GABARAP proteins were obtained based on the protocols described elsewhere [30,39]. Here, *E. coli* NEB T7

Express culture transformed with corresponding plasmids were grown till $OD_{600\ nm} = 1.0$ and protein expression was induced with 0.2 mM IPTG. The cultures were incubated at 25°C for 8–12 h before cell harvesting. Isolation and purification procedures were similar to those reported in Ref. [22,40]. Before experiments, all proteins and peptides were equilibrated with a buffer containing 50 mM $Na_2HPO_4$, 100 mM NaCl at pH 7.0, and supplied with 5 mM protease inhibitor cocktail.

The protocol for preparation of non-labelled and $^{13}C,^{15}N$-labelled PLEKHM1-LIR peptide was slightly modified to achieve highest yield of the peptide. The 50 ml M9 culture was inoculated with NEB T7 cell transformed with pET39_Ub63-PLEKHM1-LIR plasmid and grown overnight at 37°. The collected cells were resuspended in 2 l of either LB or M9 media contained 1.5 g $^{15}N$-labelled $NH_4Cl$ and 3.0 g of $^{13}C$-labelled glucose. The cultures were grown at 37° till A (600 nm) = 0.9 and supplied with 1 mM of IPTG to induce Ub63-PLEKHM1-LIR overexpression (3 h at 37°). After that cells were harvested by centrifugation, re-suspended in buffer contained 50 mM Tris–HCl pH = 7.9, 200 mM NaCl, 5% glycerol, 0.1 mg/ml DNase A and 4 mM protease inhibitor cocktail. After cell lysis by French press, debris was removed by centrifugation and clear supernatant was applied onto the column contained Ni-NTA Sepharose equilibrated with the loading buffer (50 mM Tris–HCl pH = 7.9, 250 mM NaCl, 1% glycerol and 20 mM imidazole). Elution was performed with 400 mM imidazole in the same buffer. An aliquot of pure Ub63-PLEKHM1-LIR fractions was further purified by gel filtration on Superdex75 26 × 60 column for control ITC and analytical size exclusion chromatography experiments, remaining fusion protein was processed with the TEV-protease and PLEKHM1-LIR peptide was purified to 95% purity by reverse Ni-NTA chromatography and followed gel filtration on Superdex75 26 × 60 column. Peak maximum of peptide was detected at 97 ml (void volume 115 ml). Pure peptide was concentrated in Amicon concentrators with cut-off of 3 kDa (> 95% retention).

## Crystallization and data processing

The PLEKHM1$^{629–638}$-LC3A$^{2–121}$, PLEKHM1$^{629–638}$-GABARAP$^{2–117}$ and PLEKHM1$^{629–638}$-GABARAP-L1$^{2–117}$ chimeric proteins were purified and crystallized as N-terminally LIR-fused chimeric proteins. The LC3C$^{8–125}$ protein was co-crystallized with the PLEKHM1-LIR peptide (GAMG-P$^{629}$QQEDEWVNVQYPD$^{642}$). Initial crystallization trial was performed using Hampton Research (Crystal screen, Crystal screen cryo, Index and PEG/Ion) and Molecular dimension (JCSG +, Midas, Morpheus, PACT, Clear Screen Strategy 1 and Clear Screen Strategy 1). In all cases, the drops included 400 nl of protein (concentrations listed below) and 400 nl of mother liquor. All crystallization experiments were set up at 4°C.

For PLEKHM1$^{629–638}$-LC3A$^{2–121}$ (10 mg ml$^{-1}$), crystals were grown in the JCSGplus screen condition H7 (0.2 M ammonium acetate, 0.1 M Bis Tris, pH 5.5, 25% w/v polyethylene glycol 3,350). Crystals for PLEKHM1$^{629–638}$-GABARAP$^{2–117}$ (9.1 mg ml$^{-1}$) were grown in the PEG/ion screen condition F5 (4% v/v Tacsimate pH 8.0, 12% w/v polyethylene glycol 3,350). Crystals for the PLEKHM1$^{629–638}$-GABARAP-L1$^{2–117}$ protein (7.5 mg ml$^{-1}$) were formed in the PEG/ion screen condition A6 (20% w/v polyethylene glycol 3,350, 0.2 M NaCl, 8% MPD pH 7.2). The LC3C$^{8–125}$ protein (9.2 mg ml$^{-1}$) was mixed with the PLEKHM1 peptide (2.4 mg ml$^{-1}$)

in equal volume and incubated for 3 h at 4°C, prior to setting up the crystallization trays. Crystals were formed in the PEG/ion screen condition D5 (0.2 M potassium phosphate monobasic, 20% w/v polyethylene glycol 3,350). The crystals were frozen in liquid $N_2$ prior to data collection.

X-ray diffraction data were collected on the MX2 microcrystallography beamline at the Australian synchrotron (Melbourne, Australia). The data were integrated using XDS [41] and scaled using Aimless [42]. The PLEKHM1$^{629–638}$-GABARAP$^{2–117}$ and PLEKHM1$^{629–638}$-GABARAP-L1$^{2–117}$ structures were solved by molecular replacement using MOLREP [43] and search models 1GNU and 2R2Q, respectively. Phases for the PLEKHM1-LIR:LC3C co-crystal structure were estimated using PHASER [44] and the search model was 3WAM. The solved structures were refined using PHENIX.RE-FINE [45], and manual refinement was performed using COOT [46]. The images in the work were generated using PyMOL (The PyMOL Molecular Graphics System, Version 1.5.0.4 Schrödinger, LLC).

## Isothermal titration calorimetry (ITC)

All titration experiments were performed at 25°C using a VP-ITC microcalorimeter (Malvern Instruments Ltd, UK). The ITC data were analysed with the ITC-Origin 7.0 software with a "one-site" binding model. The peptides at concentrations of 0.4 mM were titrated into 0.020 mM LC3 and GABARAP proteins in 26 steps. The protein and peptide concentrations were calculated from the UV absorption at 280 nm by NanoDrop spectrophotometer (Thermo Fisher Scientific, DE, USA).

## Nuclear magnetic resonance spectroscopy

All NMR experiments were performed at 298 K on Bruker Avance spectrometers operating at proton frequencies of 500, 600 and 700 MHz. Titration experiments were performed with a 0.18 mM $^{15}N$-labelled LC3 and GABARAP protein samples to which the non-labelled PLEKHM1-LIR peptide was added stepwise until four times excess to LC3 proteins or two times excess to the GABARAP proteins. Backbone HN resonances for selected mATG8 proteins in complex with the PLEKHM1-LIR peptide were assigned using [$^{15}N$-$^1H$]-TROSY versions of 3D HNCACB experiment. For assignment of PLEKHM1-LIR peptide backbone HN resonances in complexes with the LC3 and GABARAP proteins, [$^{15}N$-$^1H$]-TROSY versions of 3D HNCACB experiment and hCcconh-TOCSY experiment were used.

## Peptide array

Biotinylated peptides (JPT, Germany) were immobilized on streptavidin-coated 96-well plates (#436014; Thermo Scientific) in 100 μl PBS containing 0.1% Tween-20 (PT) and 1% BSA (PTB) overnight on a shaker at 8°C. After three washing steps with 200 μl PT, 100 μl of 1 μM HIS6-tagged mATG fusion proteins isolated from *E. coli* in PTB was incubated with the immobilized peptides for 1 h at 8°C. After three washing steps with 200 μl PT, HIS6-ATG8 bound to peptides was detected after 1-h incubation with anti-HIS-HRP antibody (JP-A00612; Genscript; 1:5,000 in 100 μl PTB) with the help of TMB substrate Reagent Set (BD OptEIA; 75 μl). The reaction (blue coloration) was stopped by addition of 60 μl 1 M $H_3PO_4$. Samples were analysed on a Synergy H1 ELISA reader from BioTek at 450 nm.

## Immunoprecipitation

Cells (HEK293T, HeLa and *Plekhm1*$^{+/+}$ and *Plekhm1*$^{-/-}$ mouse embryonic fibroblasts) were lysed in NP-40 lysis buffer (50 mM Tris, pH 7.5, 120 mM NaCl, 1% NP-40) supplemented with Complete® protease inhibitor (Roche). Lysates were passed through a 27 G needle, centrifuged at 21,000 *g* and incubated with either anti-GFP agarose (Chromotek, gta-20), anti-RFP (Chromotek, RTA-20) or anti-PLEKHM1 (SIGMA, HPA025018) plus Protein A agarose (Roche, PROTAA-RO ROCHE), washed three times in lysis buffer and subjected to SDS–PAGE and Western blot. Anti-GFP (Santa Cruz clone B-2, sc9996), anti-FlagM2 (SIGMA, F3165), anti-p62 (ENZO, BML-PW9860), anti-LC3B (clone 5F10 Nanotools, 0231-100/LC3-5F10) and anti-GABARAP (Abcam, ab109364) were used to detect co-precipitated proteins. Peptides were generated by China peptides with HIV-Tat sequences at the N-terminal (PLEKHM1-WT LIR peptide: GRKKRRQRRR-AEEAc-KVRPQQ**EDEWVNV**QYPDQPE; PLEKHM1-Scr-LIR peptide GRKKRRQRRR-AEEAc-VQEQQEPPPVKNYDVEQWDR). For overexpression studies, PLEKHM1-Flag, GFP-mATG8s were used as described previously [19]. p3xFLAG-CMV10-hFIP200 was a gift from Noboru Mizushima (Addgene plasmid # 24300), GFP-FUNDC1 was a kind gift from Ian Ganley, University of Dundee, and pDEST-mCherry-GFP-p62/SQSTM1 was a kind gift from Terje Johansen.

## Protein databank submission

The atomic coordinates and structure factors (PDB codes 5DPR, 5DPW, 5DPS and 5DPT for complexes of PLEKHM1-LIR with LC3A, LC3C, GABARAP and GABARAP-L1, respectively) have been deposited in the Protein Data Bank, Research Collaboratory for Structural Bioinformatics, Rutgers University, New Brunswick, NJ (http://www.rcsb.org/).

**Expanded View** for this article is available online.

## Acknowledgements

We thank Natalia Rogova for help with protein and peptide sample preparation. Work of VVR, AK, FL and VD was funded by the Center for Biomolecular Magnetic Resonance (BMRZ, Frankfurt); the German Cancer Consortium (DKTK), the LOEWE program Ubiquitin Networks (Ub-Net) funded by the State of Hesse/Germany; and the SFB 1177 "Molecular and Functional Characterization of Selective Autophagy". Work in the I.D. laboratories is supported by the DFG-funded Collaborative Research Centre on Selective Autophagy (SFB 1177); by the DFG-funded Cluster of Excellence "Macromolecular Complexes" (EXC115); by the DFG-funded SPP 1580 program "Intracellular Compartments as Places of Pathogen-Host-Interactions"; by the LOEWE program Ubiquitin Networks (Ub-Net) funded by the State of Hesse/Germany; and by LOEWE Center for Gene and Cell Therapy Frankfurt (ID) and Human Frontier Science Program (HFSP RGP55) (ID). RCJD and ARC were supported in part by Ministry of Business, Innovation and Employment Contract (contract UOCX1208), the Royal Society of New Zealand Marsden Fund (contracts UOC1013 & UOC1506) and United States Army Research Laboratory and United States Army Research Office (contract/grant W911NF-11-1-0481). RCJD and HS were supported by a Japan Society for the Promotion of Science contract through the Royal Society of New Zealand (FY2012). We thank the Australian Synchrotron for beamline access. AS was supported by the Fritz Thyssen Stiftung (Az. 10.14.2.208). Work in the DGM laboratory was funded by Wellcome Trust Seed award in Science (202061/Z/16/Z) and Tenovus Scotland (T16/44).

## Author contributions

VVR prepared all samples for the structural NMR and ITC experiments, determined $K_D$ values by NMR and ITC and wrote the manuscript; FL performed NMR experiments; AK prepared mutants and performed ITC and NMR titration experiments; AS performed and analysed peptide experiments; ACR, HS, SW and RCJD performed and analysed X-ray crystallography data; DOR-S performed co-immunoprecipitation experiments and mutations. VD, ID, RCJD and DGM wrote the manuscript. DGM designed experiments and coordinated the manuscript.

## Conflict of interest

The authors declare that they have no conflict of interest.

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
