## [Review Process File · EMBO Reports]

Manuscript EMBO-2016-43587

Structural and Functional analysis of the GABARAP Interaction Motif (GIM)

Vladimir V. Rogov, Alexandra Stolz, Arvind C. Ravichandran, Diana O. Rios-Szwed, Hironori Suzuki, Andreas Kniss, Frank Löhr, Soichi Wakatsuki, Volker Dötsch, Ivan Dikic, Renwick C.J. Dobson & David G. McEwan

Corresponding authors:

David G. McEwan, Goethe University School of Medicine and University of Dundee

Renwick C. J. Dobson, University of Canterbury and University of Melbourne

Ivan Dikic, Goethe University School of Medicine

Review timeline:

Submission date:	28 October 2016
Editorial Decision:	30 November 2016
Revision received:	20 March 2017
Editorial Decision:	11 April 2017
Revision received:	18 April 2017
Editorial Decision:	27 April 2017
Revision received:	28 April 2017
Accepted:	09 May 2017

Transaction Report:

1st Editorial Decision

30 November 2016

Thank you for the submission of your research manuscript to our journal. We have now received the full set of referee reports that is copied below.

As you will see, the referees' opinions are divided. While all three referees acknowledge the technical quality of the study, referee 2 is concerned about the novelty of the findings given that earlier studies have reported on the importance of Valine/Isoleucine at the X1 position for GABARAP binding. However, upon further discussion of this point with referees 1 and 3, who emphasized that this is the first systematic analysis using most of the known LIRs at the same time, we decided to invite you to revise your manuscript with the understanding that the referee concerns (as detailed above and in their reports) must be fully addressed and their suggestions taken on board. The analysis of the interaction between FUNDC1 and the LC3 group would certainly add to the value and novelty of the study but is not an absolute prerequisite.

Please address all referee concerns in a complete point-by-point response. Acceptance of the manuscript will depend on a positive outcome of a second round of review. It is EMBO reports policy to allow a single round of revision only and acceptance or rejection of the manuscript will therefore depend on the completeness of your responses included in the next, final version of the manuscript.

REFeree REPORTS

Referee #1:

ATG8 family proteins play pivotal roles in different processes during autophagy, including the formation of the autophagosomal membrane, selection of cargos to be sequestered into the autophagosome, and fusion of the complete autophagosome with the lysosome, in which ATG8s interact with proteins exerting a wide range of functions. Previous studies revealed that ATG8s recognize the consensus motif named the LC3-interacting region (LIR) or ATG8 family-interacting motif (AIM) in those proteins. A growing number of evidences have also revealed that different ATG8 paralogs in the same organisms bind to a certain LIR/AIM with different affinities, and this has been suggested to reflect distinct roles among ATG8 paralogs during autophagy. However, how selectivity between ATG8s and LIRs/AIMs is determined is still poorly understood. Mammals have six ATG8s, which are categorized into two subfamilies, LC3 and GABARAP subfamilies. In this study, Rogov et al performed comprehensive analysis to clarify LIR/AIM-binding preference of ATG8s, leading to the identification of the W/F-V-X-V sequence as that GABARAP proteins preferentially bind compared to LC3 proteins, and named it the GABARAP interaction motif, GIM. Experimental designs are solid, and the results are clear and sufficient to draw the conclusions the authors made in the manuscript. Identification of the GIM will accelerate our understanding distinct roles of ATG8 proteins in autophagy, and moreover, molecular mechanisms and physiological significance of this biological phenomenon. I just list some minor comments as shown below.

- (1) The definitions of receptors and adaptors in this paper should be described in the Introduction, since, unlike those used here, the term adaptors have also been used as being the same as receptors in many of previous studies.
- (2) The authors should describe the reason why cells treated with Ku and CQ were used to examine the interaction between PLEKHM1 and ATG8s.
- (3) Regarding the last sentence in page 5 "This result supports a function of PLEKHM1 as an adaptor and not a receptor protein.", I cannot understand why a preferential GABARAP binding leads to this idea. A similar sentence is also found in page 7 with citing Figure 2E. In this figure, GABARAP and LC3 are shown on the convex and concave surfaces of a growing autophagosomal membrane. There is no experimental evidence to support this specific localization of these proteins on the membrane, and thus this figure is misleading.
- (4) In the experiment shown in Figure 2D, why did the authors use PLEKHM1-LIR peptides tagged with a Tat sequence, although these peptides were added to cell lysates?
- (5) In page 8, the authors describe that "the tighter packing of the two essential residues W635 and V638 into HP1 and HP2 of GABARAPs versus LC3 families (Figure 3A and 3D and Expanded View Results).", but it is unclear how we can see this in these figures.
- (6) In page 9, the authors discuss that L or I is preferential for LC3B, potentially due to the bulkier residues present in alpha-helix 3 of GABARAPs compared to LC3s. However, this L/I preference is not the case for other LC3 family proteins, and thus this argument seems not convincing.
- (7) Regarding the result that the replacement of V636 and N637 with Gly or Pro severely impaired the interaction with ATG8s, I wonder if these mutations disrupt a beta strand configuration of the sequence rather than residue requirement specific to these positions.
- (8) In the Discussion, the authors use the terms "luminal" and "cytosolic" sides to represent the concave and convex surfaces of a growing/nascent autophagosome, respectively, but this is inappropriate, since both the sides are still cytosolic in that stage.
- (9) Given the results shown in this study, I just wonder if the authors should also include I and I/L in X1 and X3 positions in the GIM, respectively.
- (10) In Figure 3E, adding the mutant names like W635A as shown in Figure 4A may increase readability.
- (11) In Figure EV7, coloring substituted residues in the sequences shown at the bottom of the graphs red may increase readability.

Referee #2:

In this manuscript Rogov and co-workers compare established LC3 interacting region (LIR) motifs to investigate how specificity towards certain mATG8 proteins is achieved. They find that amino acids in the X1 and X2 position of the classical LIR motif (-X1-X2-) influence binding to the different mATG8 family members. Their library of LIRs indicates that GABARAP-selective binders are likely to have a Valine/Isoleucine in both the X1- and -position, furthermore by introducing a Valine in the X1 position of the p62 LIR they demonstrate that this indeed can increase the affinity for GABARAP in a LIR where Valine is absent in this position. The LIR peptide of PLEKHM1 was crystalized together with different mATG8 proteins and further analyzed for mATG8 binding specificities. Most of the data in this manuscript is clearly presented and of high quality, but there is little novelty to the findings. Several compilations of LIRs have previously been published and Alemu et al demonstrated the importance of Valine/Isoleucine in the X1 position of the GABARAP-selective binders ULK1 and ATG13 in 2012. Furthermore, the LIR of PLEKHM1 was identified and its mATG8 binding selectivity was shown by McEwan in 2015.

The interesting findings of this manuscript, in the reviewer's opinion, are related to the increased affinity achieved towards LC3. If the authors could expand on this finding the manuscript would be of a more novel nature. Like the authors state "Out of the 30 LIRs tested.....only one LIR, FUNDC1, preferentially interacted with the LC3 group". The advance given to the description of the GABARAP-selective LIR, although nicely performed, yields little advance to the field of autophagy.

The authors should also consider the following comments before submission to another journal:

- Figure 1: There is a size difference between GFP-LC3C in figure 2B compared to 2C.
- Figure 4: In figure 4C the authors control for binding of GFP to WT-PLEKHM1 but to neither of the PLEKHM1 mutants.

Referee #3:

A high number of autophagy receptors and adaptors have been reported, however their selective interaction with individual ATG8s members remains unclear. In the present study Rogov et al determine the selectivity of different LIR motifs towards LC3s and GABARAPs subfamilies. The authors screened 30 peptides derived from known autophagy receptors. Out of the 30 sequences tested, 12 shown preferential to GABARAP subfamily. By using biophysical and structural approaches the authors demonstrated that the PLEKHM1- LIR has a preference for binding to GABARAPs over LC3s subfamily. The authors also showed for the first time the crystal structures of PLEKHM1-LIR in complex with the LC3s and GABARAPs family. Finally, the model was challenged by mutating the LIR motif of p62 to prefer GABARAP over LC3B by substitute the amino acids responsible for highly interaction to GABARAP. In summary, this study provides new and interesting insights on understanding the molecular interaction of adaptors proteins to the ATG8s family and the role of ATG8s relevant for autophagosomes biogenesis, selection of cargo and fusion with the lysosome.

Specific comments:

Fig. 2B- In the IP analysis, there is a band in the same high of PLEKHM1- Flag, however no transfection with PLEKHM1 is indicated. The authors should clarify this.

Fig.2C- A control without Ku-0063794+CQ treatment is missing.

Fig.4E-D- p62 LIR alteration was used to confirm the change in LC3B to GABARAP interaction, by mutations in the core motif. However, the results shown in Fig.1A indicate that even without any mutation p62 LIR interacted slightly better with GABARAP than to LC3B. The authors should clarify this in the results and discussion sections.

Point by point response shown on the following pages:

Response to Referee's comments:

Referee #1:

(1) *The definitions of receptors and adaptors in this paper should be described in the Introduction, since, unlike those used here, the term adaptors have also been used as being the same as receptors in many of previous studies.*

Response: We have clarified the definitions of autophagy adaptors and receptors as per the reviewer's suggestion. Please see page 3 of the manuscript (red text).

(2) *The authors should describe the reason why cells treated with Ku and CQ were used to examine the interaction between PLEKHM1 and ATG8s.*

Response: Briefly, we have shown previously in the Plekhm1 Mol. Cell paper (2015) that Plekhm1 localizes to autolysosomes after stimulation with Ku+CQ. Therefore, we only used conditions of increased flux/blocked degradation in order to maximize our chances of capturing Plekhm1/mATG8 interactions. However, more importantly, we show in Figure 2D that under non-stimulated conditions (DMSO) endogenous Plekhm1 does not interact with endogenous LC3 or GABARAP proteins, but only when stimulated with Ku+CQ. We hope this explanation is acceptable for the reviewers. We have clarified this in the text on pages 6-7 (Red text).

(3) *Regarding the last sentence in page 5 "This result supports a function of PLEKHM1 as an adaptor and not a receptor protein.", I cannot understand why a preferential GABARAP binding leads to this idea. A similar sentence is also found in page 7 with citing Figure 2E. In this figure, GABARAP and LC3 are shown on the convex and concave surfaces of a growing autophagosomal membrane. There is no experimental evidence to support this specific localization of these proteins on the membrane, and thus this figure is misleading.*

Response: We agree with the reviewer that this is purely speculative at this stage and have removed the figure and references to Figure 2E.

(4) *In the experiment shown in Figure 2D, why did the authors use PLEKHM1-LIR peptides tagged with a Tat sequence, although these peptides were added to cell lysates?*

Response: The peptides we had synthesized were Tat-tagged for a different set of experiments but were used to probe this interaction. As the referee has correctly pointed out, we incubated these directly in the lysates during the IP and not added to the cells prior to lysis. We have amended the Figure 2D to remove the Tat reference to avoid confusion, but have kept the correct sequence in the materials and methods section.

(5) *In page 8, the authors describe that "the tighter packing of the two essential residues W635 and V638 into HP1 and HP2 of GABARAPs versus LC3 families (Figure 3A and 3D and Expanded View Results).", but it is unclear how we can see this in these figures.*

Response: Figure 3A and 3D has been edited with the shorter bond distances in GABARAP and GABARAPL1 represented with red lines (instead of black dashed lines for LC3s). In addition, and of more use, we have also generated a table of these bond distances and highlighted the shorter bonding distances (Appendix Table S3) and now refer to this in the text.

(6) *In page 9, the authors discuss that L or I is preferential for LC3B, potentially due to the bulkier residues present in alpha-helix 3 of GABARAPs compared to LC3s. However, this L/I preference is not the case for other LC3 family proteins, and thus this argument seems not convincing.*

Response: We agree with the reviewer and have altered the text accordingly. Now the paragraph reads: "Through substitution of W635 and V638 for residues found in other LIR sequences, we show that W635F and W635Y mutants weaken the interaction with all six

mATG8s (**Expanded View Figure EV5A**). Interestingly, V638L or V638I substitutions do not affect the interactions of PLEKHM1-LIR to the GABARAP family or LC3A and LC3C proteins, but did increase the affinity of the interaction in LC3B (**Expanded View Figure EV5A**). Overall, W635 and V638 act as the corner stones for LIR-mATG8 interaction, where the large aromatic W side chain is optimal for all mATG8s, but where the HP2 pocket that binds V638 is able to accommodate slightly larger (extra methyl) I or L residues, perhaps due to the additional conformational flexibility of the I and L side chains compared to V.”

(7) Regarding the result that the replacement of V636 and N637 with Gly or Pro severely impaired the interaction with ATG8s, I wonder if these mutations disrupt a beta strand configuration of the sequence rather than residue requirement specific to these positions.

Response: We agree with the reviewer. Indeed the role of residues at positions X_1 and X_2 should be analyzed with respect to their ability to adopt a β -conformation and participate in formation of intermolecular β -sheet. Therefore we tested this ability by prediction of secondary structure elements within a relatively short stretch of PLEKHM1 residues (26-mers, PLEKHM1 residues 625-650, wild type and mutated sequences) by JPRED4. All but one mutated sequences (V636P) maintained β -conformation, therefore we can state in the Main manuscript text (page 14) that “The effect of these substitutions is mediated by specific sidechain structure, orientation and mobility, and not by ability of mutated PLEKHM1-LIR adopts β -stranded conformation” The detailed results of the secondary structure prediction and their discussion are placed in a new section “Analysis of secondary structure elements for mutated in X_1 and X_2 positions PLEKHM1-LIR.” in Appendix (page 10).

In addition, we agree that this is possible, especially for the Pro substitution. Referring to Figure EV5 C and D, although substitution of V636 with either G or P clearly attenuates the interaction (Figure EV5 C), the same substitutions for N637 have little effect (Figure EV5 D). This can be rationalized by the observation that the backbone of V636 hydrogen bonds to the backbone of F52/F58/Y49 across both the LC3 and GABARAP families (Figure 3B): thus, an A or P substitution may well break these mainchain interactions. In contrast, the backbone of N637 faces away from the LC3/GABARAP interface and does not make any mainchain interaction (Figure 3C): thus, a P or A substitution may be more tolerated in the 637 position.

(8) *In the Discussion, the authors use the terms "luminal" and "cytosolic" sides to represent the concave and convex surfaces of a growing/nascent autophagosome, respectively, but this is inappropriate, since both the sides are still cytosolic in that stage.*

Response: We agree with the reviewer that this is the incorrect terminology to use and have replaced with concave/convex surfaces in the text.

(9) Given the results shown in this study, I just wonder if the authors should also include I and I/L in X_1 and X_3 positions in the GIM, respectively.

Response: 1) In regard to the X_3 position, we explored the effect in the context of PLEKHM1 by mutating this to I and L. This resulted in a small decrease in the interaction between Plekhh1-LIR and GABARAP proteins (Figure EV5A). However, what was clear when mutating the LIRs of p62 (WTHL), FUNDC1 (YEVL) and FIP200(FETI), the inclusion of the V in the X_1 position only slightly increased the interaction of the proteins with GABARAP (Figure 5). However, only upon mutating the I/L in position X_3 to V in combination with V in X_1 position did we observe a large increase in GABARAP interaction. Therefore, we believe that for full interaction with GABARAP, V in position X_3 is the most preferred option. Therefore, the GIM should reflect this and in this instance, we would exclude the I/L from position X_3 in our W-V- x_2 -V motif, but respect that I/L may be present in other GIMs and still show preferential interaction with GABARAP. This is reflected in Figure 1B.

2) Regarding the X_1 position, we did not extensively test the permutations in this position. However, from our data in Figure EV5C, I in position X_1 does not have much

effect on GABARAP interaction, but L in position X1 reduces GABARAP-L1 interaction. Based on the sequence analysis of all GABARAP interactors (Figure 1B), it is clear that I is also preferential.

Therefore, we have amended our consensus sequence to take this into consideration. It now reads:

[W/F]-[V/I]-x₂-V

(10) In Figure 3E, adding the mutant names like W635A as shown in Figure 4A may increase readability.

Response: We agree with the reviewer and have amended the Figures accordingly.

(11) In Figure EV7, coloring substituted residues in the sequences shown at the bottom of the graphs red may increase readability.

Response: We agree with the reviewer and have amended the Figures accordingly.

Referee #2:

Figure 1: There is a size difference between GFP-LC3C in figure 2B compared to figure 2C.

Response: The size difference in between the two gels was due to Figure 2B being run on a 7% gel and figure 2C being run on a 10% gel (due to inclusion of GFP). However, to avoid confusion we have repeated this and have replaced Figure 2B where we ran the samples on a 10% gel.

Figure 4: In figure 4C the authors control for binding of GFP to WT-*PLEKHM1* but to neither of the *PLEKHM1* mutants.

Response: We agree with the referee that these controls should have now been included and as expected, we show no interaction with GFP alone, weak interaction with of *Plekhh1*-WT with GFP-LC3B, increased interaction of *Plekhh1*-WCIL with GFP-LC3B and decreased interaction with *Plekhh1*-mutLIR.

Referee #3:

Fig. 2B- In the IP analysis, there is a band in the same high of *PLEKHM1*- Flag, however no transfection with *PLEKHM1* is indicated. The authors should clarify this.

Response: The referee was correct to point this out. We have checked our notes and at this stage we were having issues with our anti-Flag antibody and so we used anti-*plekhh1*. The faint band that the referee mentions is therefore most likely endogenous *Plekhh1*. This figure has been amended to reflect this.

Fig.2C- A control without Ku-0063794+CQ treatment is missing.

Response: Reviewer 1 also raised this point regarding this Figure (See Reviewer 1, point 2). At this stage, we wanted to maximize the potential interaction as in our 2015 Mol Cell paper, we found that *Plekhh1* localized with mATG8s after Ku+CQ. For the purposes of this experiment using overexpressed mATG8 proteins we wanted to maximize the capture of the interaction. Importantly, we did include the vehicle control in the endogenous IPs (Figure 2D) that shows no interaction with LC3 or GABARAPs under non-stimulated conditions.

Fig.4E-D- p62 LIR alteration was used to confirm the change in LC3B to GABARAP interaction, by mutations in the core motif. However, the results shown in Fig.1A indicate that even without any mutation p62 LIR interacted slightly better with GABARAP than to LC3B. The authors should clarify this in the results and discussion sections.

Response: We agree with the reviewer and have added a section in the discussion regarding this. (page 14-15, red text). It now reads:

“Interestingly, the p62/SQSTM LIR shows slight preference for GABARAP in isolated LIR-peptide assays. However, when immunoprecipitated, p62/QSSTM1 clearly shows a preference for LC3 interaction (Figure 5A). It is unclear why this may be the case, but could be due, in part, to its ability to dimerize through its PB1 domain, resulting in a conformation that is preferential for LC3 over GABARAP in cells.”

Thank you for the submission of your revised manuscript to EMBO reports. We have now received the full set of referee reports that is copied below. As you will see, both referees are very positive about the study and support publication in EMBO reports. From the editorial side, there are a few things that we need before we can proceed with the official acceptance of your study:

- You have specified 4 corresponding authors. We prefer to have either a single or up to maximally three corresponding author(s) due to ownership and responsibility issues but also to ease and simplify the communication between readers and authors. If you however have compelling reasons to share the corresponding authorship in this way, we can discuss this issue further.
- Please provide antibody information in the author checklist. In order to ease the reproducibility of the results it is also advisable to give the antibody catalog number in the Materials and Methods section.
- Please move the paragraph "Protein Databank Submission" to Materials and Methods
- Please edit the Appendix as follows: remove page 1 (title) and integrate the results into the main manuscript. In case some of the results/analysis is of very specialized interest, it is also possible to provide an EV text file, but preferentially all results should be described in the main text.
- Please provide a running title of max 40 characters and a conflict of interest statement in article.

Source data:

- Please split the source data file into one file per figure.
- Was the blot for Fig 2D first cut in half, incubated with different antibodies and then reassembled before analysis with the ChemiDOC system? If yes, please indicate the splicing with a stippled line or alike in the source data file.
- Figure 4C: In the IP with the WVNV sample there is a signal at 25 kDa (GFP). Is GFP-LC3B unstable or is this due to an overflow from the GFP control, lane 3? The same is true for Fig. 2B, where the signal for GFP-mATG8 at 37 kDa spreads into the control lanes (1, 2, GFP only) and the signal for GFP (25 kDa) spreads into lanes 3-6. Please clarify. Were the gels overloaded or are some of the fusion proteins unstable?

Please contact me any time if you have any questions. I look forward to seeing a revised version of your manuscript when it is ready.

REFeree REPORTS

Referee #1: The authors have satisfactorily addressed the comments I raised in the review of the original manuscript.

Referee #3: The authors addressed my comments and the manuscript in its present form meets EMBOR scientific merit.

We have now edited the manuscript according to your editorial suggestions and submitted the Manuscript via the online system. Below is listed a summary of the changes/responses.

- You have specified 4 corresponding authors. We prefer to have either a single or up to maximally three corresponding author(s) due to ownership and responsibility issues but also to ease and simplify the communication between readers and authors. If you however have compelling reasons to share the corresponding authorship in this way, we can discuss this issue further.

Prof. Volker Doetsch agreed to no longer be corresponding, allowing us to reduce the number to 3 (Myself, Dr. Dobson & Prof. Dikic)

- Please provide antibody information in the author checklist. In order to ease the reproducibility of

the results it is also advisable to give the antibody catalog number in the Materials and Methods section.

These are now included in Materials & Methods and author checklist

- Please move the paragraph "Protein Databank Submission" to Materials and Methods

Done.

- Please edit the Appendix as follows: remove page 1 (title) and integrate the results into the main manuscript. In case some of the results/analysis is of very specialized interest, it is also possible to provide an EV text file, but preferentially all results should be described in the main text.

As discussed, we have submitted this as part of the EV text file as we feel it is of a more specialized nature and would greatly increase the man manuscript text limit.

- Please provide a running title of max 40 characters and a conflict of interest statement in article.

"Defining a GABARAP Interaction motif" is included in main manuscript text

- Source data:

-) Please split the source data file into one file per figure.

Done

-) Was the blot for Fig 2D first cut in half, incubated with different antibodies and then reassembled before analysis with the ChemiDOC system? If yes, please indicate the splicing with a stippled line or alike in the source data file.

I have indicted the splice points with a dashed line

-) Figure 4C: In the IP with the WNVV sample there is a signal at 25 kDa (GFP). Is GFP-LC3B unstable or is this due to an overflow from the GFP control, lane 3? The same is true for Fig. 2B, where the signal for GFP-mATG8 at 37 kDa spreads into the control lanes (1, 2, GFP only) and the signal for GFP (25 kDa) spreads into lanes 3-6. Please clarify. Were the gels overloaded or are some of the fusion proteins unstable?

IN figure 2B, the GFP signal in lanes 3-6 is most likely due to a partial degradation of the LC3A and LC3B isoforms in the lysosome achieved by overexpression. This has been shown previously to happen with LC3 and ATG8 isoforms where free GFP is generated by degradation (see Klionsky et al. Guidelines PMID:26799652. Figure 8)The LC3 in control lanes is most likely due to slight overloading.

3rd Editorial Decision

27 April 2017

Thank you for the submission of your revised manuscript to EMBO reports. I apologize for the delay in getting back to you, but I have meanwhile gone through the manuscript and the changes.

I am writing with an 'accept in principle' decision, which means that I will be happy to accept your manuscript for publication once a few minor issues/corrections have been addressed, as follows.

- Thank you for reducing the number of corresponding authors to three. Please note that we need an ORCID ID for all corresponding authors and I therefore kindly ask you to provide this detail also for Dr. Dobson.

- Unfortunately, we cannot include coloured tables in the manuscript. Please provide table 1 and 2 in black & white and use symbols to highlight the AA changes.

- I recommend to indicate the possible degradation products in Fig. 4C and 2B and mention them in the respective figure legends.

- I noticed that the source data for the LC3 input is missing.

- Finally, I have looked at the EV text file and I would recommend to submit it as Appendix file, since it is rather complex and still contains a lot of information and data. While we do allow EV text files that include additional results or specialized methods, I think in this case the complexity and amount of presented data exceeds the limitations of an EV file. You don't have to change much. Please re-label the content as follows in the Appendix and when referring to it in the text. Appendix Supplementary results, Appendix table SX, Appendix Figure SX, Appendix references. Upload this file as single pdf.

If all remaining corrections have been attended to, you will then receive an official decision letter from the journal accepting your manuscript for publication in the next available issue of EMBO reports. This letter will also include details of the further steps you need to take for the prompt inclusion of your manuscript in our next available issue. Thank you for your contribution to EMBO reports.

3rd Revision - authors' response

28 April 2017

The authors made the requested changes and submitted the final version of their manuscript.

4th Editorial Decision

09 May 2017

I am very pleased to accept your manuscript for publication in the next available issue of EMBO reports. Thank you for your contribution to our journal.

YOU MUST COMPLETE ALL CELLS WITH A PINK BACKGROUND

Corresponding Author Name: David.McEwan
Journal Submitted to: EMBO Reports
Manuscript Number: EMBOR-2016-43587V2